# Dynamic regimes of the Greenland Ice Sheet emerging from interacting melt-elevation and glacial isostatic adjustment feedbacks

Maria Zeitz[1,2], Jan M. Haacker[1,2,3], Jonathan F. Donges[1,4], Torsten Albrecht[1], and Ricarda Winkelmann[1,2]

[1]Earth System Analysis, Potsdam Institute for Climate Impact Research, Member of the Leibniz Association, Telegrafenberg A31, 14473 Potsdam, Germany
[2]Institute for Physics and Astronomy, University of Potsdam, Potsdam, Germany
[3]Delft University of Technology, Faculty of Civil Engineering and Geosciences, Department of Geoscience and Remote Sensing, Delft, The Netherlands
[4]Stockholm Resilience Centre, Stockholm University, Kräftriket 2B, 11419 Stockholm, Sweden

**Correspondence:** Maria Zeitz (maria.zeitz@pik-potsdam.de), Ricarda Winkelmann (ricarda.winkelmann@pik-potsdam.de)

**Abstract.** The stability of the Greenland Ice Sheet under global warming is governed by a number of dynamic processes and interacting feedback mechanisms in the ice sheet, atmosphere and solid Earth. Here we study the long-term effects due to the interplay of the competing melt-elevation and glacial isostatic adjustment (GIA) feedbacks for different temperature step forcing experiments with a coupled ice-sheet and solid-Earth model. Our model results show that for warming levels above $2°C$, Greenland could become essentially ice-free within several millennia, mainly as a result of surface melting and acceleration of ice flow. These ice losses are mitigated, however, in some cases with strong GIA feedback even promoting an incomplete recovery of the Greenland ice volume. We further explore the full-factorial parameter space determining the relative strengths of the two feedbacks: Our findings suggest distinct dynamic regimes of the Greenland Ice Sheets on the route to destabilization under global warming – from incomplete recovery, via quasi-periodic oscillations in ice volume to ice-sheet collapse. In the incomplete recovery regime, the initial ice loss due to warming is essentially reversed within 50,000 years and the ice volume stabilizes at 61-93% of the present-day volume. For certain combinations of temperature increase, atmospheric lapse rate and mantle viscosity, the interaction of the GIA feedback and the melt-elevation feedback leads to self-sustained, long-term oscillations in ice-sheet volume with oscillation periods between 74 and over 300 thousand years and oscillation amplitudes between 15-70% of present-day ice volume. This oscillatory regime reveals a possible mode of internal climatic variability in the Earth system on time scales on the order of 100,000 years that may be excited by or synchronized with orbital forcing or interact with glacial cycles and other slow modes of variability. Our findings are not meant as scenario-based near-term projections of ice losses but rather providing insight into of the feedback loops governing the "deep future" and, thus, long-term resilience of the Greenland Ice Sheet.

## 1 Introduction

The Greenland Ice Sheet (GrIS) holds enough water to raise global sea levels by more than 7.4 m and is continuously losing mass at present, thereby contributing to global sea-level rise (Morlighem et al., 2017; Frederikse et al., 2020). Current mass loss

rates of 286 Gt/yr are observed, a 6-fold increase since the 1980's (Mouginot et al., 2019). While historically approximately 35 % can be attributed to a decrease in climatic mass balance and 65 % to an increase in ice discharge (Mouginot et al., 2019), the ratio has already shifted to approximately 50/50 (Mouginot et al., 2019; IMBIE Team, 2020). While it has been suggested that the Greenland Ice Sheet could become unstable beyond temperature anomalies of $1.6 - 3.2 \,°C$ due to the self-
amplifying melt-elevation feedback (Levermann and Winkelmann, 2016), recent studies debate whether a tipping point might have already been crossed (Robinson et al., 2012; Winkelmann et al., 2011; Boers and Rypdal, 2021). Understanding the feedback mechanisms and involved time scales at play in GrIS mass loss dynamics is necessary to understanding its stability under climatic changes.

    Changing climatic conditions during the glacial cycles had a strong influence on the ice volume of the Greenland Ice Sheet.
It varied from 3-7 m sea-level equivalent (that is the volume above floatation, divided by the total ocean area) in the last interglacial (from 126 to 115 kyrs BP) to 12 m during the last glacial maximum (19-20 kyrs BP) (Vasskog et al., 2015), while the present day volume of the GrIS is 7.42 m. Various processes and feedbacks in the ice sheet, atmosphere, ocean and solid Earth governing the ice dynamics, like ice-ocean interactions, the melt-elevation feedback, and the snow-albedo feedback played an important role in past transitions from interglacial to glacial and vice versa (Denton et al., 2010; Willeit and Ganopolski, 2018;
Pico et al., 2018). In this way, the GrIS has been a key component in the emergence of glacial cycles and their implications for overall Earth system stability, as can also be analyzed from a dynamical systems point of view (Crucifix, 2012). Simple models also allow to study the "deep future", i.e. the future on time scales beyond the ethical time horizon as defined e.g. by Lenton et al. (2019), of the Greenland Ice Sheet and the Earth system and reveal that anthropogenic $CO_2$ emissions affect the climate evolution for up to 500 kyrs and can postpone the next glaciation (Talento and Ganopolski, 2021).

The influence of the bedrock uplift onto the dynamics of the Greenland Ice Sheet has been studied with self- gravitating spherical viscoelastic solid Earth models in glacial cycle simulations by e.g. Le Meur and Huybrechts (1998, 2001). A study systematically varying the isostacy parameters was performed by Zweck and Huybrechts (2005) for the last glacial cycle. However, the interaction of the negative bedrock uplift feedback and the melt-elevation feedback, has, to our knowledge, not yet been explicitly and systematically studied in the context of the Greenland Ice Sheet (Pico et al., 2018). Here we aim to
close this research gap, by systematically exploring how the feedback between solid Earth, ice, and climatic mass balance and their interactions affect the long term response of the Greenland Ice Sheet.

    Changes in ice load lead to glacial isostatic adjustment (GIA), a decrease in ice load initiates an uplift with characteristic time scales of hundreds to thousand of years (Barletta et al., 2018; Whitehouse et al., 2019). Currently observed post-glacial uplift rates in Greenland range between -5.6 mm/yr and 18 mm/yr (Adhikari et al., 2021; Wahr et al., 2001; Dietrich et al.,
2005; Schumacher et al., 2018; Khan et al., 2008). Some studies suggest that uplift rates are higher in the South East, where the Iceland hot spot has possibly passed, which can be associated with locally low viscosities in the upper mantle (Khan et al., 2016).

    The viscous bedrock response is generally assumed to be slow compared to ice losses, with characteristic response time scales of tens to hundreds of millennia. However, several studies suggest that the viscosity of the asthenosphere and the upper
mantle varies spatially and could be locally lower than previously thought (e.g. in Iceland, Patagonia, the Antarctic peninsula,

Alaska). This implies that the time scale of the viscous response to changes in ice load might be much shorter, e.g. close to tens or hundreds of years (Whitehouse et al., 2019). The elastic response component responds on an even faster time scale to changes in ice load, e.g. the 2012 extreme melt event caused a significant peak in GPS measured uplift rates (Adhikari et al., 2017). A model of the solid Earth can help to interpret the GPS measurements in order to distinguish the elastic uplift caused by recent mass losses from the delayed viscous uplift caused by the retreat of ice since the last glacial maximum, and deduce solid earth parameters like mantle viscosity and lithospere thickness (Adhikari et al., 2021; Schumacher et al., 2018).

Efforts to model the solid earth response to changes in ice load range from local one-dimensional representations of the bedrock uplift to full three-dimensional models. The ELRA-type of model represents the solid earth as an Elastic Lithosphere and a Relaxing Asthenosphere and assigs a single time constant to the relaxation response (Le Meur and Huybrechts, 1996; Zweck and Huybrechts, 2005). These models are computationally efficient and are often coupled to ice-sheet models in long-term simulations (Robinson et al., 2012). The Lingle-Clark model expands the elastic plate lithosphere with a viscous half-space and solves the equations explicitly in time (Lingle and Clark, 1985; Bueler et al., 2007). The relaxation time of the solid earth then depends on the spatial wavelength of the perturbation in ice load, as shown in Fig. A1. However, this model uses only one constant value for the mantle viscosity, it does not include vertical or horizontal variations, nor does it solve the sea-level equation including self-consistent water-load changes or the rotational state of the Earth (Farrell and Clark, 1976; Hagedoorn et al., 2007) Such a model can be expanded to include more layers, e.g. the lower mantle, and take additional model of the relaxation time spectrum into account; however, it becomes more difficult to constrain (Lau et al., 2016). One-dimensional solid earth models explicitly consider the spherical shape of the Earth instead of assuming a half space (Tosi et al., 2005; Fleming and Lambeck, 2004; Simpson et al., 2009; Lambeck et al., 2014), but they do not represent lateral variations of solid-Earth parameters. Three dimensional models, which resolve not only several layers of the vertical dimension, but include additional variability in the horizontal direction, account for the ongoing discovery of lateral variations in mantle viscosity and lithosphere thickness (Khan et al., 2016; Whitehouse, 2018; Whitehouse et al., 2006, 2019; Haeger et al., 2019; Martinec, 2000). A laterally varying 3D model can change the estimate of projected global mean sea-level rise due to an ice-sheet collapse in the West-Antarctic by up to 10% compared to a 1D model (Powell et al., 2021). Inferred values for mantle viscosities can span several orders of magnitude and therefore substantially impact the estimate of bedrock uplift rates as a response to present day ice losses (Powell et al., 2020). So far the coupling efforts between 3D solid-Earth models and physical ice-sheet models have been focused mostly on the Antarctic Ice Sheet, exploring the feedback between solid Earth and ice sheets and its potential to dampen or inhibit unstable ice sheet retreat (Gomez et al., 2013; De Boer et al., 2014; Gomez et al., 2018, 2020). Self-gravitation effects affect the stability of the grounding line (Whitehouse et al., 2019; Pollard et al., 2017) and GIA models which self-consitently solve the sea-level equation are crucial. Ongoing work focuses on the northern hemisphere, coupling for instance the Parallel Ice Sheet Model PISM to the solid-Earth model VILMA.

Similarly, modeling efforts of the climatic mass balance of the Greenland Ice Sheet range from computationally efficient temperature index models over energy balance models to sophisticated regional climate models, an overview can be found in the model intercomparison effort by Fettweis et al. (2020).

The response of the solid earth to ice loss can be part of a negative, meaning counteracting or dampening, feedback loop, called glacial isostatic adjustment (GIA) feedback, that can mitigate further ice loss. Studies focused on the GIA feedback in context of the Antarctic Ice Sheet and the Laurentide Ice Sheet suggest that the bedrock uplift can lead to a grounding line advance and therefore has a stabilizing effect on glaciers that are subjected to the marine ice sheet instability (MISI)
(Whitehouse et al., 2019; Konrad et al., 2015; Kingslake et al., 2018; Bassis et al., 2017; Barletta et al., 2018). However, to our knowledge the GIA feedback has not yet been addressed in the context of the Greenland Ice Sheet, where, in comparison to the Antarctic Ice Sheet, marine terminating glaciers contribute less to mass loss.

The feedback cycle we explore in this study is related to the self-amplifying melt-elevation feedback. The melt-elevation feedback establishes a connection between ice thickness and climatic mass balance: the lower the surface elevations the higher
are typically temperatures and associated melt rates (see also Figure 1, in particular the red arrows). An initial increase in melt thins the ice, bringing the ice surface to lower elevation. Subsequently the temperature increases and amplifies both, melt rates and ice velocities, and therefore leads to further ice loss and thinning. Once a critical thickness is reached this feedback can lead to a destabilization of the ice sheet and irreversible ice loss (Levermann et al., 2013). (A similar feedback has also been known as the small ice cap instability, assuming constant accumulation rates above an elevation $h_S$ and constant ablation rates
below this elevation. Under these conditions a small ice cap can become unstable and expand or similarly a large ice sheet can become unstable and collapse to nothing upon small changes in the parameters (Weertman, 1961).).

The instability of the of the melt-elevation feedback, as studied by Levermann and Winkelmann (2016), assumes a static bed, so that changes in ice thickness equal changes in ice surface altitude. GIA can mitigate this feedback: Due to bedrock deformation changes in ice thickness do not directly translate to changes in surface elevation. The loss of ice reduces the load
on the bedrock and allows for a bedrock uplift, damperning the melt-elevation feedback (see blue arrows in Figure 1). Due to the high viscosity of the mantle the glacial isostatic adjustment can manifest on a slower time scale than the climatic changes which cause the ice losses in the first place.

From a static point of view a compensation of approximately 1/3 of ice thickness thinning due to GIA would be expected from Archimedes' principle, given that the ice density ($\rho_i = 910\,\mathrm{kg/m^3}$) is approximately 1/3 of the asthenosphere density
($\rho_r = 3300\,\mathrm{kg/m^3}$). In this study, we explore how the dynamic interaction of the feedbacks allows the GIA feedback not only to dampen but to (periodically) overcompensate for the melt-elevation feedback. Here we focus on the long-term stability of the Greenland Ice Sheet and how it is affected by the positive melt-elevation feedback on the one hand and the negative GIA feedback on the other hand. We use simple representations of both, the melt-elevation and the GIA feedbacks, to study the interplay between them: The melt-elevation feedback is represented by an atmospheric temperature lapse rate which affects the
melt rates. The GIA feedback is represented by the Lingle-Clark model, a generalization of the flat earth Elastic Lithosphere Relaxing Asthenosphere (ELRA) model (Bueler et al., 2007). The Lingle-Clark model accounts for non-local effects and different relaxation times depending on the spatial extent of the perturbation. We explore the parametric uncertainty range by varying the key parameters: asthenosphere viscosity for the bedrock uplift and the atmospheric lapse-rate for the melt-elevation feedback.

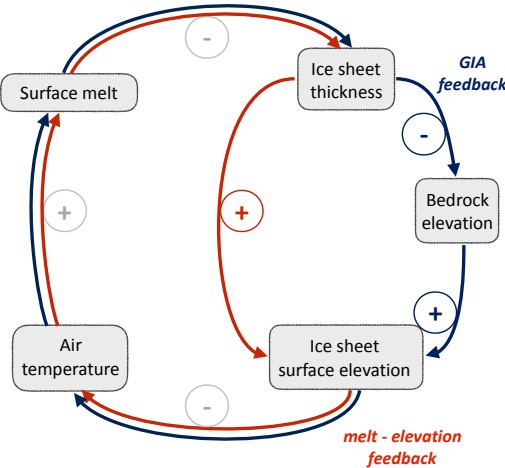

**Figure 1. Interacting feedback loops for the proposed glacial isostatic adjustment feedback (GIA feedback).** The red part indicates the melt-elevation feedback: Higher air temperatures lead to decreasing climatic mass balance. This in turn leads to a decreasing ice thickness and in consequence to a decreasing ice surface elevation. If the surface elevation decreases, the air temperature rises due to the atmospheric lapse rate. This further decreases the climatic mass balance and leads to a positive (enhancing) feedback cycle. The timescale of this feedback cycle is driven by changes in the climate and is typically comparably fast. The blue part shows the counteracting mechanism of the ice load-bedrock uplift feedback. The decreasing ice thickness reduces the load on the bedrock, which leads to isostatic adjustment and therefore an uplift of the bedrock elevation. This mechanism partly counteracts the decrease in ice surface elevation and thus mitigates further increase in temperature. The timescale of this feedback loop depends on the rate of ice retreat and on the viscosity of the upper Earth mantle.

We use the Parallel Ice Sheet Model (PISM) (Khroulev and the PISM authors, 2021; Bueler and Brown, 2009) coupled to the Lingle-Clark solid-Earth model (Bueler et al., 2007) in order to explore the interaction between the self-amplifying and the dampening feedback. The models and the experimental design are presented in Section 2. The warming experiments use an idealized temperature forcing and are analyzed in Section 3, followed by a discussion in Section 4.

## 2 Methods

### 2.1 Numerical modeling

#### 2.1.1 Ice-sheet dynamics with the Parallel Ice Sheet Model PISM

The Parallel Ice Sheet Model, PISM is a thermomechanically coupled finite difference ice-sheet model which combines the shallow-ice approximation (SIA) in regions of slow-flowing ice and the shallow-shelf approximation (SSA) of the Stokes flow stress balance in ice streams and ice shelves (Bueler and Brown, 2009; Winkelmann et al., 2011; the PISM authors, 2018). The internal deformation of ice is described by Glen's flow law, here the flow exponents $n_{\text{SSA}} = 3$ and $n_{\text{SIA}} = 3$ are used. We use

the enhancement factors $E_{\text{SSA}} = 1$ and $E_{\text{SIA}} = 1.5$ for the SSA and the SIA stress balance respectively. Different enhancement factors of the shallow ice and the shallow shelf approximations of the stress balance can be used to reflect anisotropy of the ice, as shown by Ma et al. (2010). Aschwanden et al. (2016) used in their high resolution simulations $E_{\text{SSA}} = 1$ and $E_{\text{SIA}} = 1.25$, as it provided good agreement with observed flow speeds.

The sliding is described by a pseudo-plastic power law, relating the basal stress $\tau_b$ to the yield stress $\tau_c$ as

$$\tau_b = -\tau_c \frac{\mathbf{u}}{u_{\text{threshold}}^q |\mathbf{u}|^{1-q}}, \tag{1}$$

with the sliding velocity $\mathbf{u}$, the sliding exponent $q = 0.6$ and the threshold velocity $u_{\text{threshold}} = 100 \, \text{m/yr}$. The yield stress is determined from parameterized till material properties and the effective pressure of the saturated till via the Mohr-Coulomb criterion (Bueler and van Pelt, 2015):

$$\tau_c = (\tan \phi) N_{\text{till}} \tag{2}$$

with the till friction angle $\phi$, linearly interpolated at the beginning of the run from the bedrock topography between $\phi_{\text{min}} = 5°$ and $\phi_{\text{max}} = 40°$ between bedrock elevations of -700 m and 700 m. The effective pressure on the till $N_{\text{till}}$ is determined from the ice overburden pressure and the till saturation as described in Bueler and van Pelt (2015).

### 2.1.2    Earth deformation model

While global GIA models with sea-level coupling are available, to our knowledge no coupling efforts between ice-dynamics and solid-Earth models have been undertaken for the Greenland Ice Sheet specifically. Here, the deformation of the bedrock in response to changing ice load is described with the Lingle-Clark (LC) model (Lingle and Clark, 1985; Bueler et al., 2007), incorporated as solid-earth module in PISM. In this model the response time of the bed topography depends on the wavelength of the load perturbation for a given asthenosphere viscosity (Bueler et al., 2007). The LC model uses two layers to model
the solid earth: the viscous mantle is approximated by a half space of viscosity $\eta$ and density $\rho_r$, complemented by an elastic layer of flexural rigidity $D$ describing the lithosphere. The response of the elastic lithosphere happens instantaneously, while the response time of the viscous mantle lies between decades and tens of millennia, depending on both, the viscosity of the mantle and the wavelength of the change in load. While the Lingle-Clark model is not considering local changes to viscosity or lithosphere thickness (Milne et al., 2018; Mordret, 2018; Khan et al., 2016) and approximates the earth as a half space, the
relatively small spatial extent of the simulation region allows for such an approximation.

The resulting partial differential equation for vertical displacement $u$ of the bedrock can be described by

$$2\eta |\nabla| \frac{\partial u}{\partial t} + \rho_r g u + D \nabla^4 u = \sigma_{zz}. \tag{3}$$

with $g$ being the gravitational acceleration of the earth and $\sigma_{zz}$ the ice load force per unit area (Bueler et al., 2007).

Here the flexural rigidity $D$ is assumed to be $5 \times 10^{24} \, \text{Nm}$, assuming a thickness of 88 km for the elastic plate lithosphere
(Bueler et al., 2007). The mantle density $\rho_r$ is approximated with $3300 \, \text{kg/m}^3$.

Following Bueler et al. (2007), Eq. (14), we show how the spectrum of the relaxation time of the solid Earth model depends on the wavelength of the ice load change and how this relationship changes for different mantle viscosities and lithosphere flexural rigidities in Fig. A1. In general high mantle viscosities shift the spectrum to higher relaxation times, the maximal relaxation time increases by more than two orders of magnitude, from approx 100 yrs to approx 50000 yrs, while the thickness of the lithosphere has a less strong effect on the relaxation times spectrum.

### 2.1.3 Climatic mass balance and temperature forcing

The climatic mass balance in PISM is computed with the positive degree day (PDD) model from 2m-air temperature and precipitation given as inputs (Braithwaite, 1995). Here we use a yearly cycle of monthly averages from 1958 to 1967 of the outputs of the regional climate model RACMO v2.3 (Noël et al., 2019) in order to mimic preindustrial climate. The warming is implemented as a spatially uniform instantaneous shift in temperature. The temperature forcing itself has a yearly cycle, with the temperature shift in winter being twice as high as in summer. This corresponds to an average Arctic amplification of 150 % (see also Robinson et al. (2012)).

The PDD method uses the spatially uniform standard deviation $\sigma = 4.23$, the melt factors for snow and for ice $m_i = 8\,\mathrm{mm K^{-1} day^{-1}}, m_s = 3\,\mathrm{mm K^{-1} day^{-1}}$ (PISM default) respectively. The melt-elevation feedback is approximated by an atmospheric temperature lapse rate $\Gamma$, so that local changes in the ice-sheet topography alter the temperature as

$$T_{ij} = T_{ij,\text{input}} - \Gamma \cdot \Delta h_{ij}, \tag{4}$$

with $T_{ij}$ being the effective temperature at grid cell $i,j$ feeding into the PDD model. $T_{ij,\text{input}}$ is the temperature at $i,j$ given by the input, without any lapse rate correction, $\Gamma$ is a spatially constant air-temperature lapse rate. $\Delta h_{i,j} = h_{t,ij} - h_{0,ij}$ is the local difference in surface elevation at $i,j$ at time $t$, compared to a reference topography $h_0$. Here we use the initial state for $h_0$. The value of the lapse rate $\Gamma$, and thereby the strength of the melt-elevation feedback, is varied between 5 K/km and 7 K/km in the experiments.

The yearly precipitation cycle remains fixed and does not scale with temperature, the local temperature affects how much of the precipitation is perceived as snow and therefore adds to the accumulation: at a temperature above $2°\mathrm{C}$, all precipitation is perceived as rain, below $0°\mathrm{C}$ all is perceived as snow, with a linear interpolation between the two states (the default in PISM). The climatic mass balance is adjusted via a flux correction in the regions which are ice-free in present-day to keep them ice-free. How variations in these three assumptions affect the results is discussed in Sec. 4.3.

### 2.2 Experimental design

Here we use a spatial resolution in x and y direction of 15 km in order to do many simulations over 0.5 million years. The spatial resolution in z-direction is quadratically decreasing from 36 m in the cell closest to the bedrock to 230 m in the top grid cell of the simulation box (at 4000 m above bedrock).

The temperature forcing is a spatially uniform step forcing, which is applied from $t = 0$ over the whole simulation time. Additional local temperature changes happen due to the atmospheric temperature lapse rate, as shown above. We explore

different values for the atmospheric lapse rate in order to estimate the response of the system to changes in the strength of the melt-elevation feedback.

The ice-ocean interaction is modeled via PICO, with ocean temperatures and salinities taken from the World Ocean Atlas version 2 (Zweng et al., 2018; Locarnini et al., 2019) and remapped onto the simulation grid of 15 km horizontal resolution. PICO used one average value of temperature and salinity per extended drainage sector (for the extended sectors, the drainage sectors of Rignot and Mouginot (2012) are extended linearly into the ocean), even as the ice sheet advances or retreats. The averages are taken at bottom depth over the continental shelf. The warming signal at depth generally stabilizes at lower levels than the atmospheric or sea surface warming, here we assume that only 70% of atmospheric warming reaches the ocean ground (see also Albrecht et al. (2020)). However, only less than 0.2% of the Greenland Ice Sheet area are made up of floating ice tongues and the ocean forcing is not transferred to the ice fronts of grounded tidewater glaciers, so the ice-ocean interaction is not the main driver in this simulation setup.

Calving is modeled as a combination of eigencalving (Winkelmann et al., 2011; Levermann et al., 2012) and von-Mises calving (Morlighem et al., 2016) with constant calving parameters. In addition a maximal floating ice thickness of 50 m is imposed.

### 2.2.1 Initial state

The initial state is in equilibrium for constant climate conditions. The misfits of the initial state compared to observed velocities (Joughin et al., 2018) and thicknesses (Morlighem et al., 2017) and to modelled climatic mass balance (Noël et al., 2019) are shown in the Supplementary Material in Figures S1, S2 and S3. All simulations are run at a spatial horizontal resolution of 15 km. The basic dynamics of the melt-elevation feedback and the GIA feedback are well captured at this resolution, which allows to explore the parameter space effectively. However, a lot of features of the complex flow of outlet glaciers are not captured at this resolution.

### 2.2.2 Choice of model parameters

We chose to vary along three main parameters. On the one hand, we vary the strength of the melt-elevation feedback by varying the atmospheric temperature lapse rate $\Gamma$ between 5 K/km and 7 K/km. Many ice-sheet models use the free air moist adiabatic lapse rate (MALR), which ranges between 6-7 K/km (Gardner et al., 2009) for high humidity, but assumed to be higher in cold temperatures when the air is dry (Fausto et al., 2009). However, the mean slope lapse rates measured in Greenland and on other ice caps in the Arctic tend to be lower than the MALR and show seasonal variation (Fausto et al., 2009; Gardner et al., 2009; Steffen and Box, 2001; Hanna et al., 2005). By using spatially and temporally constant lapse rates between 5-7 K/km we try to cover a realistic range in lapse rates.

In addition, the response time and strength of the bedrock to changes in ice load is determined by the mantle viscosity $\eta$, varied between $1 \times 10^{19}\,\mathrm{Pa \cdot s}$ and $5 \times 10^{21}\,\mathrm{Pa \cdot s}$. This range is larger than the values of the upper mantle viscosity given in the literature, which still range over more than two order of magnitude over Greenland alone, usually around $1 \cdot 10^{20}\,\mathrm{Pa \cdot s}$ to $5 \cdot 10^{21}\,\mathrm{Pa \cdot s}$, but local values from $1 \cdot 10^{18}\,\mathrm{Pa \cdot s}$ to $1 \cdot 10^{23}\,\mathrm{Pa \cdot s}$ cannot be ruled out (Tosi et al., 2005; Adhikari et al., 2021;

**Table 1.** Parameters used in experiments

| Name | Parameter | Value |
|---|---|---|
| $\Delta T$ | Temperature increase | $[1.5, 2, 3]\,\text{K}$ |
| $\Gamma$ | Atmospheric temperature lapse rate | $[5, 5.5, 6, 6.5, 7]\,\text{K/km}$ |
| $\eta$ | Mantle viscosity | $[1 \times 10^{19}, 1 \times 10^{20}, 1 \times 10^{21}, 5 \times 10^{21}]\,\text{Pa}\cdot\text{s}$ |

Mordret, 2018; Khan et al., 2016; Wahr et al., 2001; Peltier and Drummong, 2008; Larour et al., 2019; Le Meur and Huybrechts, 1996, 1998; Milne et al., 2018; Fleming and Lambeck, 2004; Lecavalier et al., 2014; Lambeck et al., 2014; Lau et al., 2016). Ice retreat itself is affected by the temperature anomaly, here varied between 1.5 K and 3.0 K global warming (note the arctic amplification of 150 % leading to higher local temperature anomalies).

**3   Results**

Here, we analyze how the strengths of the melt-elevation feedback and the GIA feedback influence the long term dynamics of the Greenland Ice Sheet in PISM simulations.

**3.1   Temporal evolution of ice volume under temperature forcing depending on atmospheric lapse rate and mantle viscosity**

The ice losses in simulations with applied warming are affected by both the amplifying melt-elevation feedback and the mitigating GIA feedback. The interaction of both feedbacks allows for a variety of dynamic regimes, depending on the amount of warming on the one hand and the parameters describing the feedback strength on the other hand.

At a given temperature anomaly (here $\Delta T = 2\,\text{K}$) and a given mantle viscosity (here $\eta = 1 \times 10^{21}\,\text{Pa}\cdot\text{s}$ ), both, the rate and magnitude of the initial volume loss increase with increasing air temperature lapse rate, i.e. a stronger melt-elevation feedback

(see Figure 2 (A)). With a lapse rate of $\Gamma = 5\,\text{K/km}$, at the low end of the tested range, an incomplete recovery after an initial ice loss is observed, the ice sheet loses approx 1.5 m sea-level equivalent in volume, before stabilizing at 6 m SLE after approx. 50 kyrs (1 m SLE corresponds to approx. 361 800 Gt of ice). With an increasing lapse rate and thereby increasing strength of the melt–elevation feedback the ice volume may still recover after a stronger initial loss. At sufficiently high lapse rates the recovered state is not stable on long time scales. A self-sustained oscillation of repeated ice losses and gains is observed for

$\Gamma = 6\,\text{K/km}$ with an oscillation time scale of approx. 109 kyrs. Increasing the lapse rate even further, to $\Gamma = 7\,\text{K/km}$ does not allow the ice to recover at all, the ice volume is permanently lost.

Here, depending on the value of the lapse rate $\Gamma$ three qualitatively different response regimes are observed, (i) incomplete recovery, (ii) self-sustained quasi-periodic oscillation, and (iii) permanent ice loss.

In contrast a constant lapse rate of $\Gamma = 6\,\text{K/km}$, a warming of $\Delta T = 2\,\text{K}$ and varying mantle viscosities between $\eta = 1 \times$

$10^{19} - 5 \times 10^{21}\,\text{Pa}\cdot\text{s}$ lead to self-sustained oscillations (ii) in the ice sheet volume independently of the value of the mantle

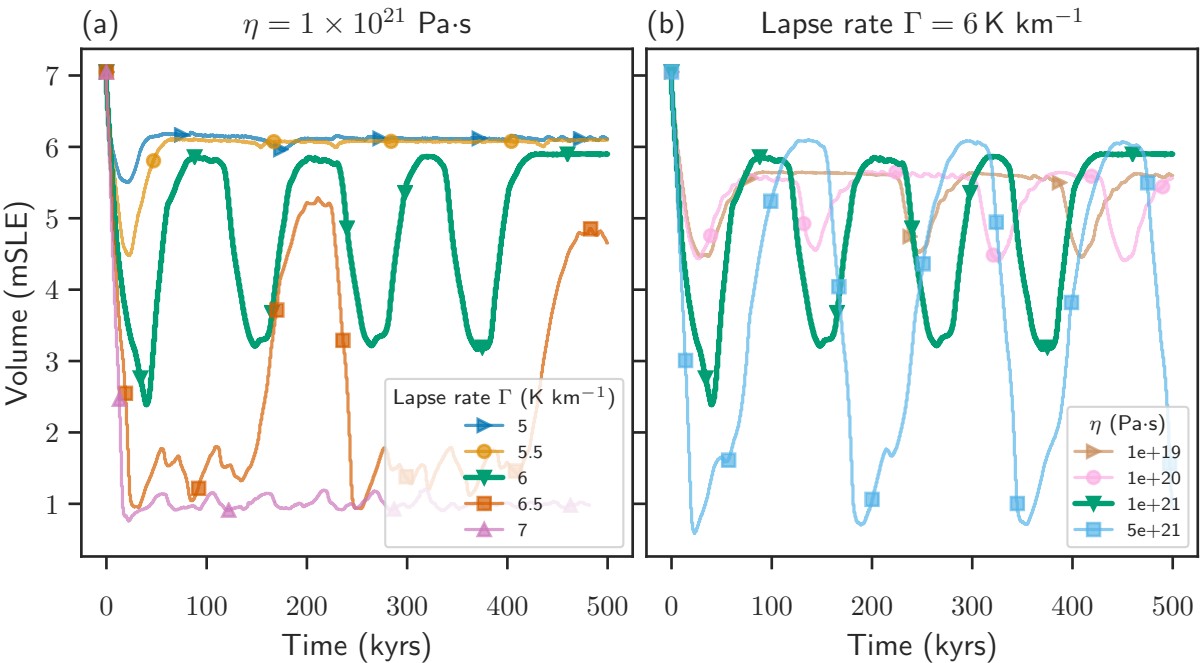

**Figure 2. Temporal evolution of Greenland Ice Sheet volume** at a temperature anomaly of $\Delta T = 2K$. Depending on the atmospheric temperature lapse rate (between 5 and 7 K/km) (a) and on the mantle viscosity (between $1 \times 10^{19}$ and $5 \times 10^{21}$ Pa·s) (b) distinct regimes of dynamic responses are observed, including incomplete recovery, quasi-periodic oscillations and permanent loss of ice volume.

viscosity (see Figure 2 (B)). The variations in mantle viscosity do not change the dynamic regime qualitatively, they affect however the time scale and the amplitude of the observed oscillations. Large values for the mantle viscosity are associated with a smaller response time scale of the GIA and thereby allow for larger initial ice losses and large amplitudes of oscillation. The amplitude, here taken as the difference between the maximal and the minimal volume after an initial ice loss, increases from

5    1.2 m sea level equivalent to 5.5 m sea level equivalent by increasing the mantle viscosity from $1 \times 10^{19}$ Pa·s to $5 \times 10^{21}$ Pa·s. 1 m sea level equivalent corresponds to approx. 361800 Gt of ice

      The spatial configuration of the ice thickness, the bedrock topography and the equilibrium line, separating the accumulation from the ablation region in response to warming is visualized for one example simulation in the oscillation regime (ii), with the parameters $\Delta T = 2\,\mathrm{K}$, $\eta = 1 \times 10^{21}\,\mathrm{Pa \cdot s}$ and $\Gamma = 6\,\mathrm{K/km}$. We choose three points in time, representing the initial state,

10   the state with minimal ice volume and the oscillation maximum, a recovered state which is unstable on long time scales (see Figure 3). The time evolution of the volume is depicted by the thick green curve in Figure 2. During the retreat phase, the mass loss of the ice is initiated from the north of the ice sheet. The area and volume of the ice sheet decrease and reach a minimal value after approx. 40 kyrs, with a remaining ice dome over central Greenland and a second smaller ice patch over the southern tip of Greenland. This ice loss is accompanied by an uplift of the bedrock which is most prominent in areas with complete ice

15   loss. The maximal ice thickness decreases from 2940 m to 2270 m in the minimal volume state, attained in the Eastern region

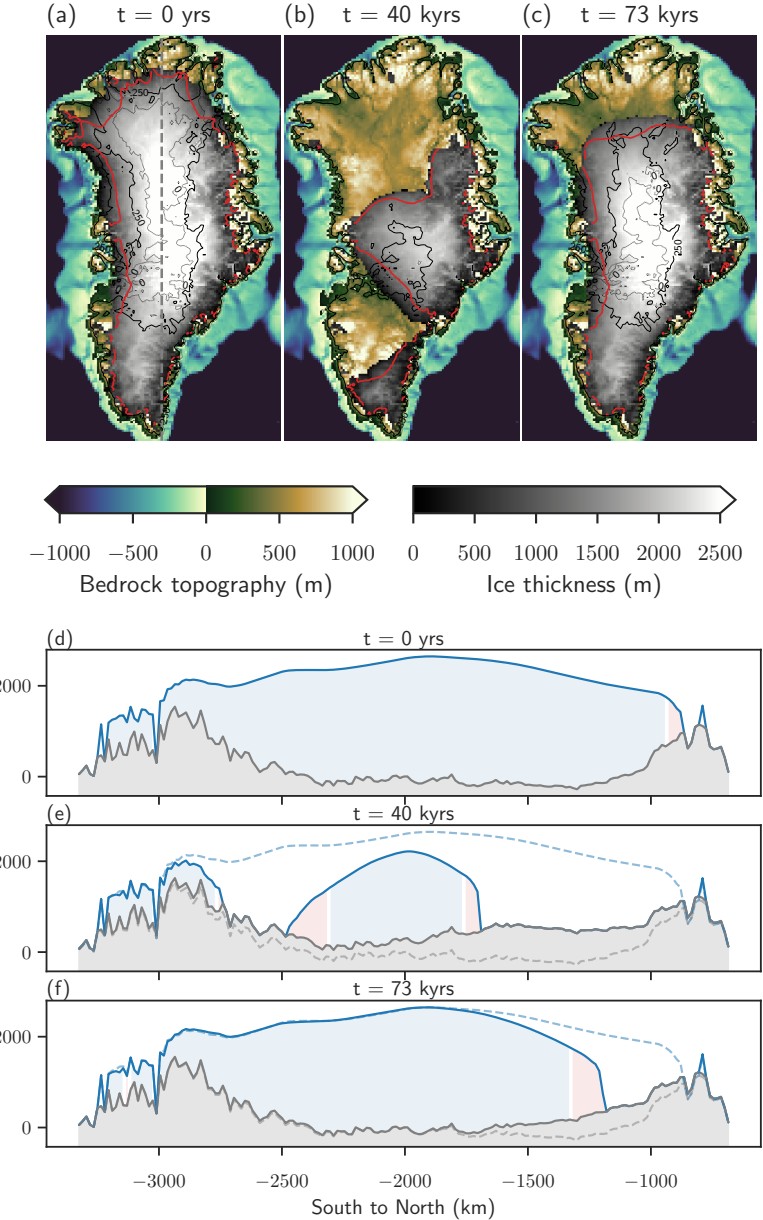

**Figure 3. Spatial distribution of ice thickness** at a temperature anomaly of $\Delta T = 2\mathrm{K}$ for the parameters $\eta = 1 \times 10^{21}\,\mathrm{Pa \cdot s}$, $\Gamma = 6\,\mathrm{K/km}$. Maps and cross-sections of the bedrock topography and ice thickness show the initial state at the start of the simulation (a,d), the state with minimal volume after 40 kyrs (b,e) and the recovered state after 73 kyrs (c,f). The red outline and the red shaded areas indicate the ablation regions. The dashed lines in (e) and (f) show the initial topography of the ice sheet.

of the ice larger ice dome. The maximal bedrock uplift of 740 m is found in the northern region where the most ice is lost. The minimal state is also characterized by an increase in relative ablation area, 29% compared to 24% in the initial state. The maximal relative ablation area of 31% is reached approx. 500 years before the minimum of the volume is reached. Eventually, the accumulation area expands and allows the ice sheet to regrow. However, the maximally recovered ice sheet differs from the initial state in area, thickness distribution, accumulation area, and bedrock topography (see Figs. 3 C and F). In particular the ice sheet extends much less to the north than in the initial state. The precipitation field is assumed to be constant in time, there is no feedback between the ice sheet topography and the precipitation pattern.

## 3.2 Competing positive melt-elevation and negative GIA feedbacks

Here we explore the competing feedbacks by varying the parameters, which determine the relative strengths of the involved feedbacks, simultaneously.

### 3.2.1 Dynamic regimes

To gain a better understanding of the dynamic regimes of the GrIS we tested the long-term response of the ice-sheet volume to warming in the full-factorial parameter space $\Delta T, \Gamma$ and $\eta$ with values given in Table 1. As stated above in Section 3.1 four qualitatively different response regimes can be distinguished: (i) incomplete recovery to a stable state after an initial ice loss. (ii) Self-sustained qasi-periodic oscillations and (iii) Irreversible loss of a large portion of the ice or (iv) direct stabilization into a new equilibrium state which preserves 90% or more of the initial ice volume. Note that only oscillations with a minimal amplitude of 0.5 mSLE are included in the oscillating regime.

With increasing temperature anomalies $\Delta T$ a larger portion of the parameter space experiences irreversible ice loss (iii) (Figure 4). For a warming temperature of 3 K for example, irreversible ice loss is observed for lapse rates greater or equal than 6 K/km for all mantle viscositites and for 5.5 K/km for mantle viscosities lower or equal to 1e+20 Pa·s.

Moreover, increasing temperature lapse rate promotes irreversible ice loss, for instance at $\Gamma = 7$ K/km, irreversible ice loss occurs for warming temperatures of 2 K or warmer, regardless of the choice for the mantle viscosity (see Figure 4) and also for most simulations with $\Delta T = 1.5$ K.

Direct stabilization without going though a minimum (o) is only realized for the lowest temperature anomaly $\Delta T = 1.5$ K and at the lowest lapse rate $\Gamma = 5$ K. While incomplete recovery or stabilization are the most prevalent regimes for low warming temperatures (1.5 K) in the tested parameters space, the oscillatory regime is realized most often at temperature anomalies of 2 K. High mantle viscosities promote oscillations of the ice sheet volume as they lead to a slower response of the bedrock to changes in ice loss and thereby allow for a stronger retreat phase and thereby a faster initial ice loss with warming, as seen in Fig. 2. On the other hand, the more pronounced retreat initiates a strong bedrock response which supports the recovery. However, the recovered state is not in equilibrium with the bedrock, and thereby a self-sustained oscillation can be triggered.

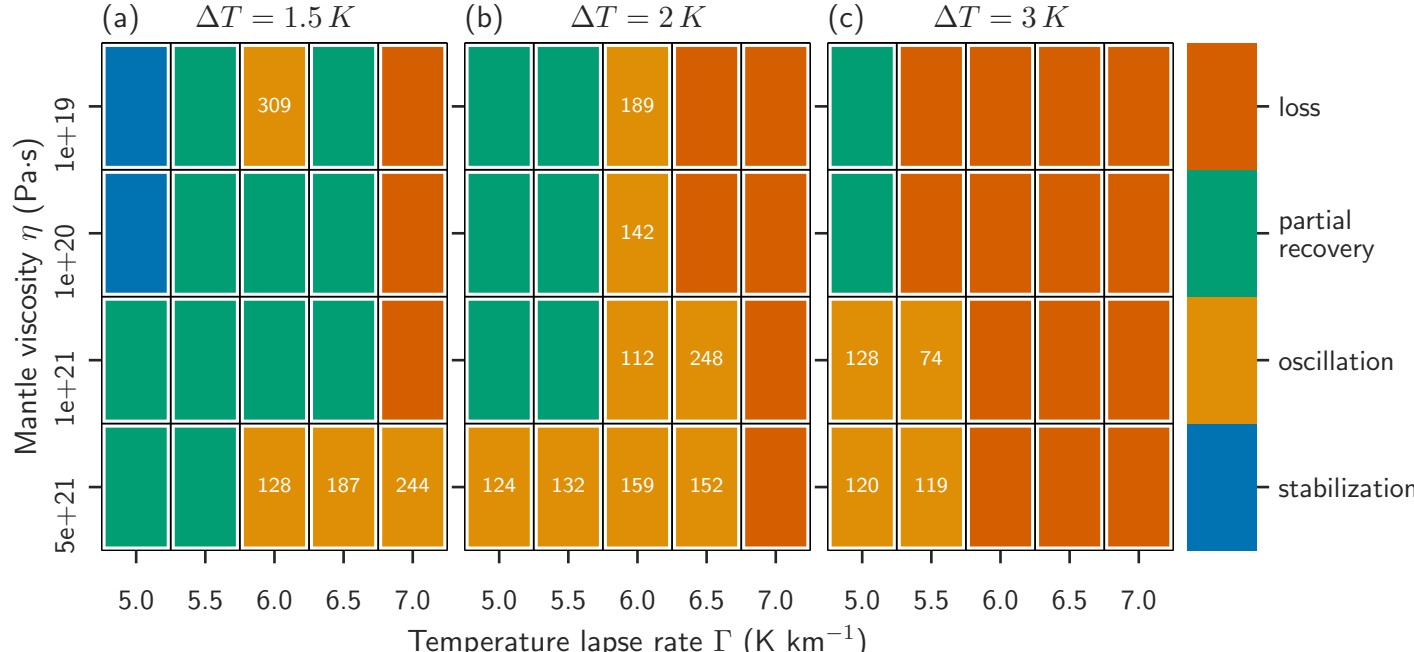

**Figure 4. Dynamical regimes of Greenland ice sheet evolution** for three different warming temperatures, $\Delta T = 1.5\,\mathrm{K}$, $2\,\mathrm{K}$, and $3\,\mathrm{K}$. The evolution regime of the Greenland Ice Sheet volume is indicated by the color code. Purple indicates immediate stabilization to a stable state, which preserves more than $90\,\%$ of the initial ice sheet volume, without passing a minimum. Green indicates, that the ice sheet volume recovers permanently after passing a minimum first. Blue indicates, that the ice sheet volume does not recover permanently, but shows self-sustained oscillations on a long time scale instead. Red indicates a permanent loss of ice sheet volume. The numbers in the cyan tiles show the approximate oscillation times.

### 3.2.2 Time scales in the oscillation regime

The observed oscillations in ice sheet volume (regime iii) are not perfectly periodical, therefore the concepts of periodicity or frequency cannot be directly applied. This framing would require that the physical state of the ice sheet, regarding not only its volume but also spatially resolved variables like thickness distribution, velocity fields, the state of the solid earth and the climate, return to exactly the same state after one oscillation period. Instead we here estimate the characteristic duration of the oscillation via a simple algorithm: first we identify the minimal and maximal volume of the oscillation, and the center of both. As the oscillation is not symmetric, the time-average of the volume would be not centered between the minimum and the maximum. In a next step we measure the time between two intersections of the time series with the central oscillation volume, which would correspond to a half oscillation, if those were perfectly symmetric and periodic. The average time between one intersection and the next but one corresponds to the oscillation time. As only a few oscillation periods fit in to the simulation time of $500\,\mathrm{kyrs}$, a thorough statistical analysis can not be performed. Note that the uncertainty arising from choosing the central

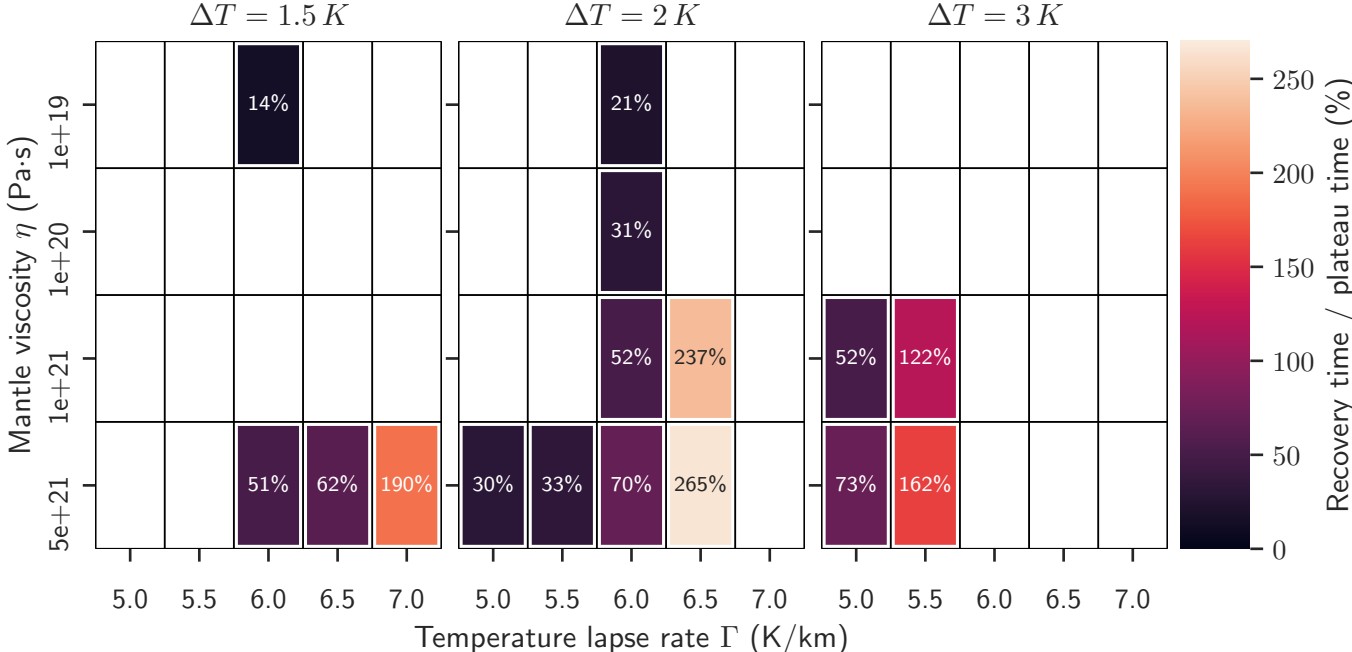

**Figure 5. Ratio of recovery time vs. plateau time** for three different warming temperatures, $\Delta T = 1.5\,\mathrm{K}$, $2\,\mathrm{K}$, and $3\,\mathrm{K}$. Measure for how long the lower half of the oscillation is compared to the upper half. A ratio of 100% would signify a perfectly symmetric oscillation, the lower the number, the more time is spent in a recovered regime (ice volume > average of minimal and maximal oscillation volume), and the higher the number the more time is spent at low ice volumes (ice volume < average of minimal and maximal oscillation volume).

volume between the minimal and the maximal volume, rather than a weighted average, are much less than the uncertainty due to the imperfect periodicity, as they amount to less than 1% in most cases and to about 2.5% in the worst case.

The oscillation times, as shown in Fig. 4, in this study vary between 79 kyrs (for $\Delta T = 3\,\mathrm{K}$, $\eta = 1 \cdot 10^{21}\,\mathrm{Pa \cdot s}$ and $\Gamma = 5.5\,\mathrm{K/km}$) and 250 kyrs (for $\Delta T = 2\,\mathrm{K}$, $\eta = 1 \cdot 10^{21}\,\mathrm{Pa \cdot s}$ and $\Gamma = 6.5\,\mathrm{K/km}$). An even longer oscillation time of 309 kyrs is

5 found for $\Delta T = 1.5\,\mathrm{K}$, $\eta = 1 \cdot 10^{19}\,\mathrm{Pa \cdot s}$ and $\Gamma = 6.0\,\mathrm{K/km}$, which is however strongly asymmetric: the ice sheet volume seems to recover and reach a permanently stable plateau, but after approx 250 kyrs a decline in ice-sheet volume is re-initiated. The oscillation times do not seem to show a clear dependence on the values for warming, lapse rate or mantle viscosity. Rather, it is governed by a more complex interplay of the dynamics: time scale and depth of the initial deglaciation, level of maximally recovered volume, stability of the plateau between ice losses.

10 The analysis method described above, allows to distinguish between the average time for the lower half of the oscillation ("recovery time") and the upper half of the oscillation ("plateau time"). We find that generally the recovery time is shorter than the plateau time, 14% in the case of $\Delta T = 1.5\,\mathrm{K}$, $\eta = 1 \cdot 10^{19}\,\mathrm{Pa \cdot s}$ and $\Gamma = 6.0\,\mathrm{K/km}$ (oscillation time 309 kyrs). This ratio increases with temperature forcing $\Delta T$, with the mantle viscosity $\eta$, and most strongly with the lapse rate $\Gamma$ up to 265% for the

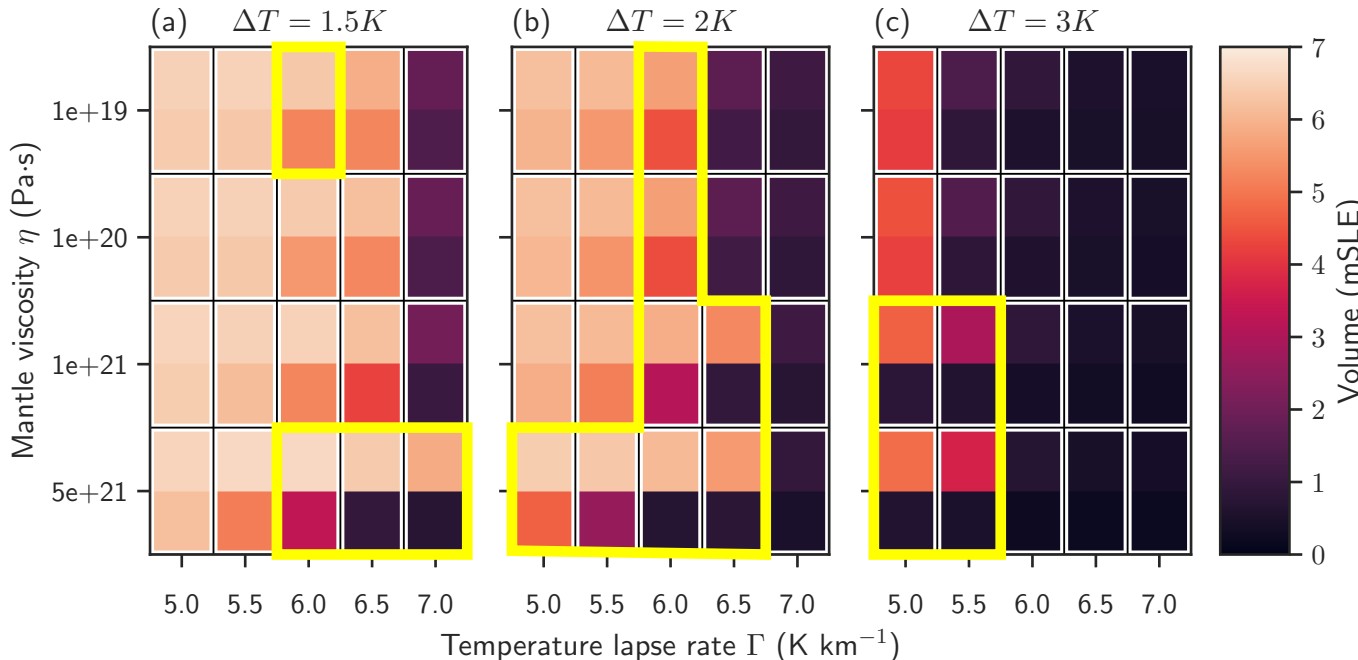

**Figure 6. Minimal and maximal volume after initial ice loss** for three different warming temperatures, $\Delta T = 1.5\,\mathrm{K}$, $2\,\mathrm{K}$, and $3\,\mathrm{K}$. Two pixels represent the minimal (lower) and the maximal (upper) long-term volume for each parameter combinations. The minimal and maximal long-term volume is defined by the minimum or the maximum of the volume after passing the initial minimum. A significant difference between the minimal and the maximal volume indicates oscillation. The yellow outline highlights the parameters space of the oscillating regime.

parameter combination of $\Delta T = 2\,\mathrm{K}$, $\eta = 5 \cdot 10^{21}\,\mathrm{Pa\cdot s}$ and $\Gamma = 6.5\,\mathrm{K/km}$ (see Fig. 5). The smaller this ratio, the more stable the partially recovered state of the ice sheet.

### 3.2.3 Minimum and maximum ice volume for incomplete recovery or oscillation regimes

The long-term response of the Greenland Ice Sheet volume to temperature anomalies can be characterized by the minimal

5    and maximal long-term volume, defined as the minimal and maximal volume attained after passing an initial minimum. In the dynamic regimes of stabilization, incomplete recovery, and permanent ice loss the minimal and maximal long-term volumes are therefore almost identical. The absolute values of the minimal and maximal long-term volume determine how much ice is lost and the difference between them shows how large the amplitude of the oscillation is. The minimal and maximal long-term volumes are visualized in Figure 6. Here, two values are shown for each parameter combination. The upper pixels represent the

10    maximum long-term volume, while the lower pixel represents the minimum long-term volume. A comparison to the regime shown in Figure 4 reveals that both volumes are high if the ice volume is stabilized directly or recovers, and both volumes are low if the ice is permanently lost. Oscillations are characterized by a significant difference between the minimal and maximal

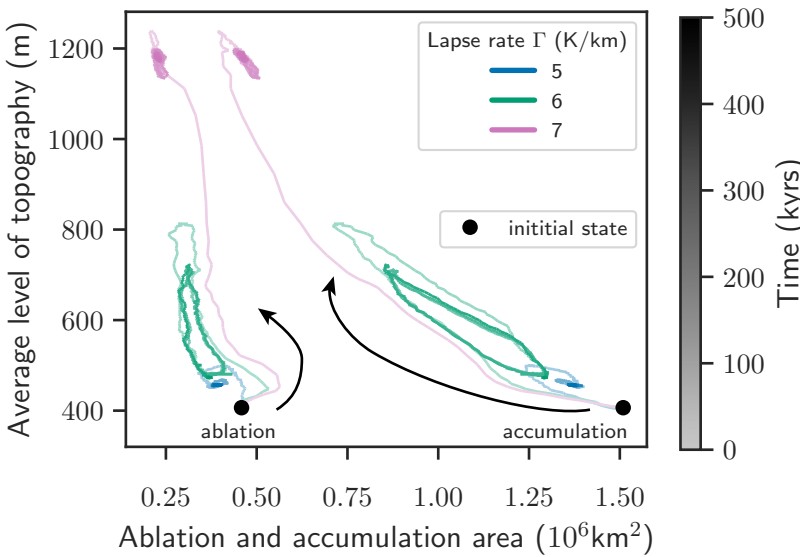

**Figure 7. State space trajectories for different regimes** for the three different lapse rates $\Gamma = [5, 6, 7]$ K/km. The curves represent the average height of the bedrock topography vs. the ablation and the accumulation area. Blue: Lapse rate $\Gamma = 5$ K/km, incomplete recovery of the ice volume. Green: Lapse rate $\Gamma = 6$ K/km, oscillation of the ice volume. Purple: Lapse rate $\Gamma = 7$ K/km, irreversible loss of the ice volume. The color code corresponds to Figure 2 a, that is the temperature forcing is $\Delta T = 2$ K and the mantle viscosity is $\eta = 1 \times 10^{21}$ Pa·s. The color shading represents time, as indicated by the color bar on the right side. The average bedrock and the accumulation and ablation area of the initial state are represented by the black marker.

long-term volume. Generally, the absolute values for either stabilized, oscillating or lost volume decrease with increasing warming. The amplitude of oscillations is highest for high mantle viscosities, since the slow response time associated with high mantle viscosities allows for more ice loss but also for a stronger recovery.

### 3.3 State space trajectories

Here we analyze the different dynamic regimes of the Greenland Ice Sheet via their trajectories through state space. The full state space of an ice sheet has a very high dimensionality and even with the simplifications made by numerical modelling, the full state space remains inaccessible. Here we choose the projection of the state to three state variables: the temporal evolution of the average topography altitude of the glaciated areas on the one hand and the ablation or accumulation area of the ice sheet on the other hand. In both cases the variables are averaged over glaciated areas rather than over a fixed area (e.g. the initial ice sheet area), because this is where they affect the ice sheet. For instance the bedrock uplift in a region which has (permanently) lost its ice does not take part in the feedback as described in Fig. 1. The average topography altitude can change either via glacial isostatic adjustment while the ice sheet area is constant or by changing the ice sheet area at constant topography or a combination of those two processes. The ablation or accumulation area can either change though changes in the ice-sheet

area at constant climatic mass balance or via changing the climatic mass balance but keeping the ice-sheet area constant (or a combination of those two processes).

We interpret the topography altitude as a measure of the GIA feedback and the accumulation and ablation area as a measure for the climatic processes.

We can distinguish three different "phase space trajectories" for the different regimes: incomplete recovery after an initial ice loss (i), oscillation (ii) and ice-sheet collapse (iii). All of the simulations shown here are at $\Delta T = 2\,\mathrm{K}$ and with the mantle viscosity of $\eta = 1 \times 10^{21}\,\mathrm{Pa \cdot s}$.

For $\Gamma = 5\,\mathrm{K/km}$ (blue curves in Figure 2 (A) and Figure 7), the ice sheet is in the incomplete recovery regime. Both the accumulation / ablation areas and the average topography diverge the least from the starting point compared to the other simulations. The trajectories for both accumulation and ablation area spiral quickly into a fixed point. The trajectory for the ablation area follows a counterclockwise spiral while the trajectory for the accumulation area follows a clockwise spiral. Here, changes in accumulation area seem to be more important than the changes in the ablation area and to drive the dynamics.

For $\Gamma = 6\,\mathrm{K/km}$ (green curves in Figure 2 (A) and Figure 7) the ice sheet is in the oscillation regime. The trajectories spiral into a closed loop rather than a fixed point, which is characteristic for limit-cycles and non-linear oscillators. Again, the trajectory with the ablation area goes counter-clockwise while the trajectory for the accumulation area goes clockwise. In absolute terms the accumulation area changes more drastically than the ablation area during one cycle, an indication that the change in accumulation area drives the ice loss. Even though these trajectories form closed loops, there is no perfect periodicity in the beginning, as the first loop of the trajectory is larger than the subsequent following ones.

The atmospheric lapse rate of $\Gamma = 7\,\mathrm{K/km}$ (purple curves in Figure 2 (A) and Figure 7) leads to irreversible ice-sheet collapse under these parameters. The trajectories approach again a fixed point. Both the accumulation and the ablation area are smallest, compared to the other two lapse rate simulations, indicating that the total area of the ice sheet is also small. Again, the absolute change in accumulation area is more drastic than the change in ablation area and the change in average level of bedrock topography is highest. As indicated beforehand, this is both related to the bedrock uplift (most ice loss allows for the strongest uplift) as well as to the fact that the remaining ice retreats to high altitude mountainous areas with a lot of precipitation and comparatively low temperatures.

## 4 Discussion

### 4.1 GIA feedback in different contexts

The impact of the GIA feedback on ice-sheet dynamics has been studied in different contexts. Marine terminating glaciers and ice shelves are particularly sensitive to glacial isostatic rebound, as it can influence the position of the grounding line and how exposed the ice shelf or the glacier front is to warm ocean water (Larour et al., 2019; Whitehouse et al., 2019). Observational evidence pointing to an overshoot and readvance of the grounding line in the Ross Sea, Antarctica, can be explained by the viscous response of the solid Earth to changes in ice load within a confined range of mantle viscosities (Kingslake et al., 2018).

Feldmann and Levermann (2017) showed, that the complex interplay of time scales associated with the surge, buildup and stabilization feedbacks could explain Heinrich-like events.

## 4.2 GrIS ice volume oscillations

While oscillations of ice volume have already been discussed in the context of marine ice sheets (Antarctic Ice Sheet, Laurentide
Ice Sheet) (Bassis et al., 2017), we here find that the interaction of the melt-elevation feedback and the GIA feedback can promote an oscillatory dynamic response in a mostly grounded ice sheet.

### 4.2.1 Analysis of oscillation times

The observed oscillation times vary widely over the range of tested parameters, between 74 kyrs and 309 kyrs (see Fig. 4). However, the asymmetric shape of the oscillations and their irregularity makes it difficult to establish a straight forward dependence
between the oscillation time itself and the parameters determining the dynamical response. When analyzing the asymmetry of the oscillations, however, a clear pattern emerges. The fraction of the time the GrIS spends during recovery in a low-volume or collapsed state compared to the time it spends in a high-volume plateau (here termed "recovery time" and "plateau time") depends strongly on the parameters (see Fig. 5). The "recovery time" fraction increases with increasing warming temperature $\Delta T$, with increasing lapse rate $\Gamma$, and with increasing mantle viscosity $\eta$. The fact, that relatively more time is spent in a low
volume state, seems to indicate a loss of stability of the Greenland Ice Sheet. This is particularly true for the warming temperature and the lapse rate, as they also promote the transition from the oscillatory regime to the collapse of the Greenland Ice Sheet. Although high mantle viscosities $\eta$ promote the oscillatory regime, they also allow for higher ice loss rates and higher total amounts of ice loss, and therefore destabilize the ice sheet in our simulations.

### 4.2.2 GrIS ice volume oscillations in the context of the Earth System

The oscillation times, even if irregular, are of the same order of magnitude as the time scale of Earth's glaciation cycle, with a dominant period of 41 kyrs before and a period of 100 kys after the Mid-Pleistocene Transition 1.25–0.7 million years ago (Abe-Ouchi et al., 2013; Willeit et al., 2019). While the onset and the termination of glaciation are driven by changes in insolation, climate and earth surface albedo (Ganopolski and Calov, 2011) our results offer a new perspective. The identified oscillatory regime reveals a possible mode of internal climatic variability in the Earth system on time scales on the order of
100 kyrs that may be excited by or interact with orbital forcing, glacial cycles and other slow modes of variability (Ghil and Lucarini, 2020). As such, this oscillatory mode could be relevant in the long-term Earth system response (on the order of 100 kyrs) to anthropogenic carbon emissions (Talento and Ganopolski, 2021).

Our findings suggest a sequence of dynamic regimes of the Greenland Ice Sheets on the route to destabilization under global warming, within a certain range of lapse rate coefficients: from recovery, via quasi-periodic variations in ice volume to
irreversible ice-sheet collapse. This transition might be similar to destabilization scenarios via oscillatory instabilities which have been revealed for other tipping elements in the climate system, such as the Atlantic Meridional Overturning Circulation

(AMOC) (Alkhayuon et al., 2019). A relevant area of future research will be to develop a deeper understanding of such ice sheet destabilization routes via the concept of bifurcations (e.g., Hopf and fold bifurcations) in the context of dynamical systems. The interplay between an amplifying and a mitigating feedback contributes to our understanding of the long-term stability and the resilience of the Greenland Ice Sheet. Therefore we need to identify the most important underlying physical processes and the interactions of the feedbacks at play.

## 4.3 Robustness analysis

While large amplitude oscillations generated with a process based ice-sheet model have not been reported in the peer-reviewed literature, small oscillations in the GrIS ice volume seem to appear in simulations with the CISM ice-sheet model coupled to an ELRA bedrock model (Petrini et al., 2021) under constant climate. Although the oscillatory regime is not studied explicitly by Petrini et al., its appearance indicates that this dynamic regime is unlikely to be an artifact of our particular experimental design.

In addition Oerlemans (1982) found unforced oscillations in a simple ice-sheet model, including simple representations of the melt-elevation feedback (depending on the latitude as well as on the altitude), the thermodynamics of the ice sheet including sliding and the bedrock uplift (using a constant relaxation time). They have found thermodynamics to be necessary for the appearance of oscillations. Even though the amplitude and period found by Oerlemans (1982) are very sensitive to parameter choice, the free oscillations seem to be a robust feature of that model over a wide range of parameter values, confirming that the interaction of both feedback shown in Fig. 1 can indeed generate robust oscillation.

In order to make sure that the observed dynamical regimes discussed in the present study, in particular the oscillating regime, are not an artifact produced by specific modeling choices, we perform several robustness checks. In the following the impact of some assumptions made for the bedrock model and for the climatic mass balance is tested for one set of parameters ($\Gamma = 6\,\text{K/km}$, $\Delta T = 2\,\text{K}$, $\eta = 1 \times 10^{21}\,\text{Pa}\cdot\text{s}$.).

Changing the bedrock uplift model to the instantaneous point-wise isostacy model, defined as

$$b(t,x,y) = b(0,x,y) - \frac{\rho_i}{\rho_m}\left[H(t,x,y) - H(0,x,y)\right],$$

and leaving all other parameters and modeling choices fixed produces very similar oscillations to the reference run (see Fig. 8). Recovering an oscillating regime with instantaneous isostacy shows that the time lag between ice load change and full uplift is not the only driver of the oscillation. The change in bedrock model causes a decrease in amplitude, by shifting the minimal volume up, and an increase in oscillation time.

Including the precipitation scaling of 7.3% per degree Celsius of global mean temperature change, in contrast to the fixed precipitation field, mitigates the ice losses and leads to higher minimal and maximal volumes and a decrease in oscillation amplitude and an increase in oscillation time.

In order to test the impact of the initial state, a different spin-up was performed in addition to the equilibrium spin-up, which was used for the standard runs. Here we use a spin-up similar to the "paleo-climate spin-up" in Aschwanden et al. (2013) over the last 125 kyrs. However, the simulation of the past 125 kyrs including bedrock deformation is performed twice, adding

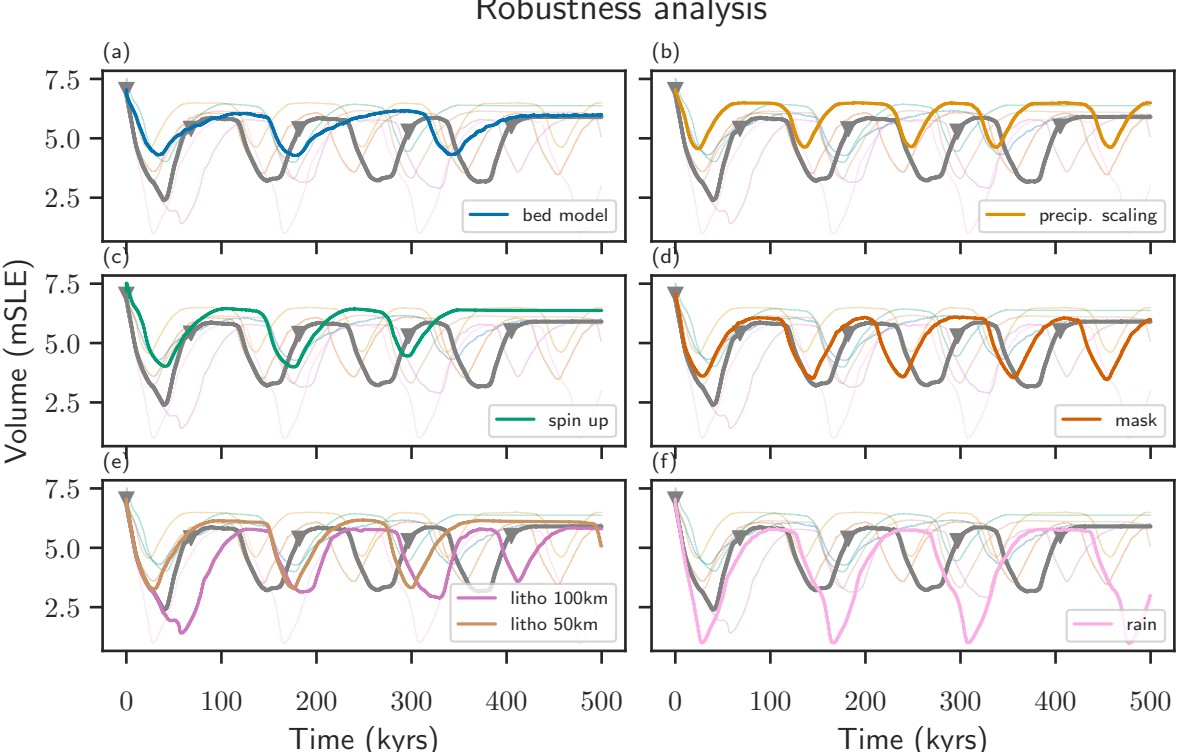

**Figure 8. Robustness analysis** for the simulation run with parameters $\Gamma = 6\,\text{K/km}$, $\Delta T = 2\,\text{K}$, $\eta = 1 \times 10^{21}\,\text{Pa·s}$. The gray curve corresponds to the reference run, also shown in Fig. 2 and 3 and is shown in each panel for reference. The colored faint lines provide context. Each change in modeling choice is highlighted in its own panel. (a) Run with a instantaneous pointwise isostacy model. (b) Run which includes a 7.3% precipitation increase per degree Celsius of global mean temperature increase. (c) Run which starts from a glacial spin-up. (d) Run which omits the flux correction at the ice-free margin. (e) Runs with two different lithosphere thicknesses, 100 km and 50 km. (f) Run, which uses a different interpolation between rain and snow.

the anomaly of the bedrock topography at the end of the first run to the initial state of the second run. Therefore an initial state closer to present day topography is obtained at the end of the second run, and the bedrock is in equilibrium with the ice topography. Using this paleo-climate spin up with explicit treatment of the bedrock still recovers the oscillatory regime for the first 300 kyrs (see Fig. 8 c). In contrast to the reference run, the amplitude of the oscillation decreases with time and a stable

5   plateau is observed in the past 150 kyrs.

In the reference run we adapted the climatic mass balance in the areas which are ice-free under present day in order to keep them ice-free, such that the initial state would not grow beyond the area of present day ice sheet. The flux correction at the ice free margins has only a minor effect on the oscillating regime (see Fig. 8). The oscillation amplitude is barely altered, the

oscillation time is slightly shorter, and the initial ice loss is less deep compared to the reference run. However, the volume of the unforced control run grows from 7.06 m to 7.62 m.

The influence of the lithosphere thickness, which can be altered indirectly through a different flexural rigidity of the lithosphere, which is proportional to the third power of the lithosphere thickness. Increasing the lithosphere thickness from 88 km to 100 km increases the initial ice loss and the oscillation time, however the long-term amplitude of the oscillation and the minimal and maximal volume remain almost unaffected. Decreasing the lithosphere thickness to 50 km reduces the initial ice loss and increases the maximal volume of the oscillation. An almost stable plateau of approx. 150 kyrs appears after 350 kyrs of simulation time, but a dip in the ice volume is observed at the end of the oscillation time, indicating that the plateau is not stable on the long term.

In the reference run all precipitation is perceived as snow if the local mean temperature is below 0°C and all precipitation is perceived as rain if the local mean temperature is above 2°C, with a linear interpolation in between. Changing the critical temperatures to -1°C and 3°C allows a bigger window where both, rain and snow are present. This change introduces a larger oscillation amplitude and reduces the oscillation time (see Fig. 8 f).

The modeling choices will most likely also affect the distribution of the dynamical regimes in the parameter landscape as shown in Fig. 4, and changing more than one modeling choice at one would introduce stronger changes. For instance changing the spin-up and flux correction (see Fig. 8, c and d) at once shifts the regime from "oscillation" to "incomplete recovery". Recreating simulations for the full parameter space for each of the modeling choices and different combinations of those is, unfortunately, beyond the scope of this paper. However, we have shown that the oscillating regimes of the Greenland Ice Sheet under constant temperature forcing are robust against many modeling choices, including first tests with PISM interactively coupled to the global VIscoelastic Lithosphere and MAntle model (VILMA, see Klemann et al. (2008); Martinec et al. (2018)) in forthcoming work, and is therefore unlikely to be an artifact created by one particular simulation setup.

## 4.4 Limitations

This study is based on the results of the ice sheet model PISM coupled to simple models which capture the melt-elevation feedback, namely the positive degree day approach together with an atmospheric temperature lapse rate, and the GIA feedback, namely the Lingle-Clark model. The relative computational efficiency of those models allows us to conduct an ensemble of long-term simulations over 500,000 years exploring different parameter values characterizing the individual feedbacks and warming. This approach fits the conceptual research question of this study.

The Lingle-Clark approach assumes a flat earth with two layers, one elastic and one viscous layer, in contrast to more sophisticated solid Earth models. It also does assume horizontally constant Earth structure and does not solve the self-consistent sea-level equation. However, the relative importance of discharge and melt at the ice-ocean interface decreases with ongoing warming, as the tidewater glaciers retreat and the ice-ocean interface shrinks (Aschwanden et al., 2019). With ongoing coupling efforts between ice-dynamics models and process based solid Earth models, this study is a first step to assessing the importance of the GIA feedback for the stability of the Greenland Ice Sheet.

While the design of the study was chosen in order to allow for long experiments and to cover parts of the parameter space ($\Delta T$, $\Gamma$, and $\eta$), it is also one of the main limitations of the study. The coarse resolution of the ice sheet model does not adequately resolve the flow patterns in outlet glaciers, therefore underestimating dynamical ice losses. Moreover, the parameters which govern the ice dynamics, although uncertain, were not varied (Zeitz et al., 2020).

The choice of the positive degree day (PDD) method in order to compute the climatic mass balance introduces a tends to underestimate the melt area for present climate, while at high temperatures PDD tends to overestimate melt. Moreover, the temperature anomaly is applied in a spatially and temporally constant way and the precipitation pattern remains fixed, without any adaption to the temperature forcing. A more realistic approach would include the increase of precipitation with warmer air temperature and partially mitigate ice losses. However in this rather conceptual study we explore the stability landscape without taking the increase in precipitation into account, as it reduces the complexity of the system. We have shown in 4.3 that while the total ice losses are reduced when considering the increase in precipitation, the qualitative dynamics remains the same (oscillations). So far, scenario-based projections of future global warming are limited until the year 2300, with projections of the temperature evolution and changes in climatic mass balance over the Greenland Ice Sheet as results from regional climate models only available until the end of this century. The aim here, however, is not to present scenario-based projections of future ice losses but rather to study the distinct dynamical states in the "deep future" of the Greenland Ice Sheet in a fundamental way.

## 5 Conclusions

Here we present an analysis of the dynamic regimes in the deep future of the Greenland Ice Sheet. Depending on the amount of warming and the values of the parameters describing the strength of the melt-elevation feedback and the GIA feedback we find that four different dynamic regimes can be realized: 1) Direct stabilization into a new equilibrium state which preserves 90% or more of the initial ice volume, 2) Incomplete recovery to a stable state after an initial ice loss, 3) Self-sustained oscillations, and 4) Irreversible loss of a large portion of the ice. Our model configuration with parameterized melt-elevation feedback and a fast computation of the leading-order GIA effects allows for studying an ensemble of glacial time-scale simulations and provides insight into how the interaction of feedbacks impacts the dynamics of the complex Earth system with implications for Earth system stability and resilience. Although it is not explicitly studied here, drastic changes in the ice volume of the Greenland Ice Sheet would have implications for the global earth system via global sea level rise, changes in the planetary albedo, and changes in the atmospheric and oceanic circulation patterns as the Jetstream or the Atlantic Meridional Overturning Circulation (AMOC).

*Code availability.* The PISM source code is freely available.

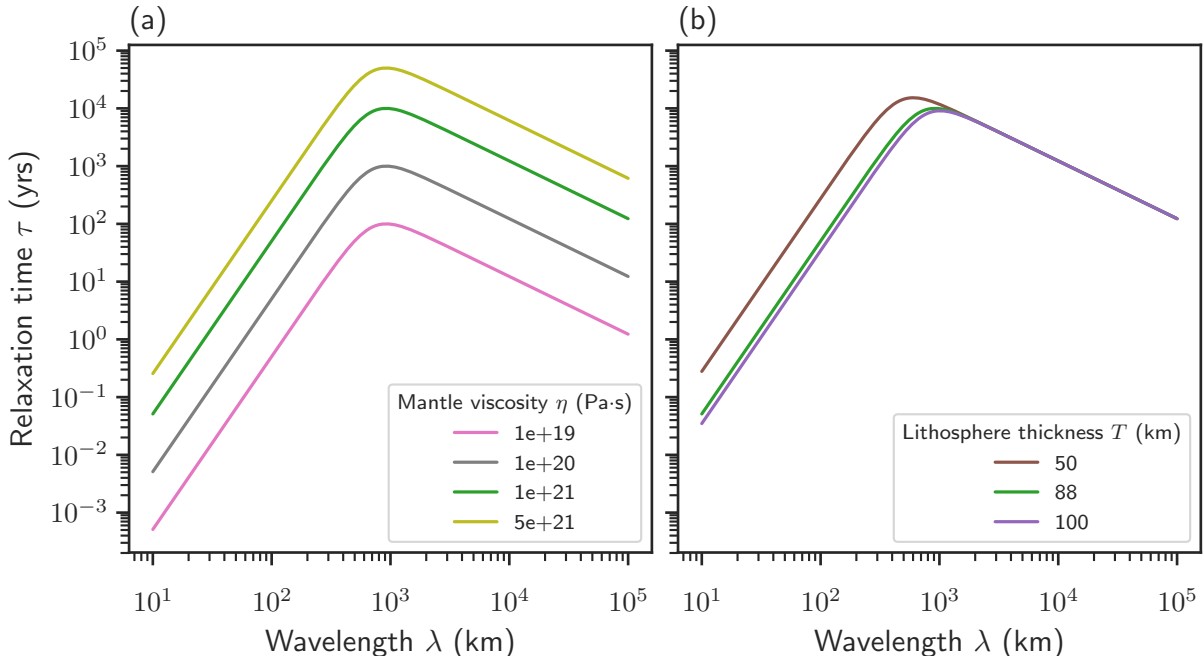

**Figure A1.** Spectrum of the relaxation time vs. wavelength of the load change for different mantle viscosities and lithosphere thicknesses, as shown in Bueler et al. (2007), Eq. (14)

## Appendix A: Relaxation times in the Lingle-Clark model

Following Bueler et al. (2007), Eq. (14), the relaxation time of the Lingle-Clark model can be computed as a function of the load change wavelength $\lambda$ from comparison to the ELRA model as

$$\tau(\lambda) = \frac{4\pi\eta|\lambda|^{-1}}{\rho_m g + 16\pi^4 D\lambda^{-4}}.$$

5   As the relaxation time is directly proportional to the mantle viscosity $\eta$ the maximal relaxation time increases by more than two orders of magnitude over the tested parameter range. The changes in lithosphere thickness induce less changes to the relaxation time spectrum. Wavelengths relevant for the deglaciation of the Greenland Ice Sheet are between several tens of kilometers (onset of retreat) to 500-1500 km (the spatial extent of the Greenland Ice Sheet).

*Author contributions.* RW conceived this study. JH prepared and analysed the initial experiments during his Master project, advised by RW
10   and MZ. MZ expanded the experiments, visualized the results and wrote the manuscript, with support from RW and JFD and TA. All authors interpreted and discussed the results. All the authors give their final approval of the article version to be published.

*Competing interests.* The authors declare no competing interests.

*Acknowledgements.* We would like to thank the editor, Michel Crucifix for handling our manuscript and Kristin Poinar and one anonymous reviewer for their helpful comments on the initial version of the manuscript. Moreover, we would like to thank Michele Petrini, Anders Levermann and Fuyuki Saito for fruitful discussions . We are grateful for financial support by the European Research Council Advanced Grant project ERA (Earth Resilience in the Anthropocene, grant ERC-2016-ADG-743080) and the Leibniz Association (project DominoES). MZ is partially supported by the Deutsche Forschungsgemeinschaft (DFG) by grant WI4556/3-1. MZ acknowledges the German Fulbright commission for a PhD program. RW and TA acknowledge support by TiPACCs, PROTECT, PalMod. We further acknowledge the European Regional Development Fund (ERDF), the German Federal Ministry of Education and Research (BMBF) and the Land Brandenburg for supporting this project by providing resources on the high-performance computer system at the Potsdam Institute for Climate Impact Research. Development of PISM is supported by NASA grant NNX17AG65G and NSF grants PLR-1603799 and PLR-1644277.

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
