# Peer review of "Dynamic regimes of the Greenland Ice Sheet emerging from interacting melt-elevation and glacial isostatic adjustment feedbacks"

_Earth System Dynamics, 2021_

## Author Comment (AC2)

**Response to Reviewer Comments**

Journal: Earth System Dynamics

Title: **Dynamic regimes of the Greenland Ice Sheet emerging from interacting melt-elevation and glacial isostatic adjustment feedbacks**

Authors: Maria Zeitz, Jan Haacker, Jonathan Donges, Torsten Albrecht and Ricarda Winkelmann

MS No: esd-2021-100

MS Type: Research article

First of all, we would like to thank the editor Michel Crucifix and the two reviewers, Kristin Poinar and one anonymous reviewer, for their immensely helpful comments and their efforts to create the detailed reviews! In our revision of the manuscript we addressed the main issues:

1. In order to address the question, if the observed behavior is an artifact, we have now included an additional robustness analysis, including a different initial state and a different solid Earth model. The qualitative behavior remains the same.

2. We have added a more thorough discussion of the Lingle-Clark model

3. We have renamed the dynamic regimes and added a more detailed discussion on the fluctuation times.

We provide detailed answers to all comments below. The reviewers' comments are given in black and the authors' in blue. The changes made to the manuscript can be found at the end of this document (created with latexdiff). In addition to the changes suggested by the reviewers, we have changed the variable name for the mantle viscosity from $\nu$ to $\eta$, to be more consistent with existing literature, e.g. Bueler et al. (2007), and we have improved Figure 1 visually (without changing its content).

**Comment on esd-2021-100**

Kristin Poinar (Referee)

Referee comment on "Dynamic regimes of the Greenland Ice Sheet emerging from interacting melt-elevation and glacial isostatic adjustment feedbacks" by Maria Zeitz et al., Earth Syst. Dynam. Discuss., https://doi.org/10.5194/esd-2021-100-RC1, 2022

**Summary and general comments:**

This manuscript presents a discovery of unforced, long-term fluctuations in the size of the Greenland Ice Sheet. The fluctuations (which are not really oscillations, as they are not strictly regular or repeating) have periods ~80 - 300 kyr and originate from the interactions between the melt-elevation feedback (a positive feedback) and glacial isostatic adjustment (a negative feedback). This has not been previously studied on long (ice age) timescales in the absence of external triggers (e.g., Heinrich events initiated by ocean heat pulses) for a land-terminating ice sheet. The finding of these emergent cycles could be relevant for "deep future" states of the Greenland Ice Sheet, although it is a challenge to imagine a future without climate forcings that would presumably overshadow the internal variability. Regardless, it is an interesting discovery

that merits reporting, and this paper is largely successful. I have only minor suggestions, and although they are somewhat numerous, they are all quite attainable.

Many thanks for the review! We understand that imagining a long-term stable climate forcing may seem somehow abstract. Nevertheless we believe that understanding the dynamic response of the GrIS to this comparatively simple forcing lays an important foundation, e.g. for understanding the stability of the Greenland Ice Sheet.

Specific comments:

The authors used a "power spectrum analysis" to identify periods in the ice volume time series. These methods should be explained, if only briefly, and some test for significance should be carried out. The authors state that "The oscillation times do not seem to show a clear dependence on the values for warming, lapse rate or mantle viscosity" (P11 L12). This seems troubling -- wouldn't we expect a clear pattern to emerge within the parameter space? If so, the authors should do additional thinking and put forth possible explanations for the scatter. If not, that is interesting too, and the authors should elaborate on the reasons why this system is not governed regularly.

Many thanks for this comment, which inspired us to look into the oscillation times more closely. As they do not seem to be perfectly regular or symmetric, showing e.g. extended plateaus at high ice volumes with rather brief "dips" in volume (see e.g. gray and pink curve in Figure 2 b), and only a few oscillations fit into the simulation time of 500kyrs, we now chose a slightly simpler approach to analyze the time scales involved. In particular, we identify the average of the minimal and maximal oscillation volume and analyze the times between two intersections of the ice volume curve with that value. Thus we do not only get the oscillation time for a full period (which we have updated), but we can also analyze the half oscillations more carefully. In some cases the time between two "dips" is very long and therefore dominates the oscillation time. We have separated both time scales, the "recovery time" (defined as the average time for the half oscillation going through a minimum, black lines in Figure R1) and the "plateau time" (defined as the average time for a half oscillation going through a maximum, gray lines in Figure R1) .

[Figure]

*Figure R1: Illustration of recovery time and plateau time. The oscillation of the ice volume is divided into the half oscillation going through the minimum (black lines) and through the half oscillation going through the maximum (gray lines). The volume time series is taken from Figure 2 in the manuscript, with parameters $\Delta T = 2K, \Gamma = 6K/km, \eta = 1 \times 10^{21}Pa\,s$.*

We discuss the complex interactions between the recovery time, the depth of the dip, the maximal recovered ice, or the amplitude of oscillation very briefly in the text of the revised manuscript. More importantly, we now analyze the ratio of the recovery time to plateau time. Here a clear pattern emerges: the upper half of the oscillation gets relatively shorter with increasing temperature forcing, increasing mantle viscosity, and most importantly with increasing lapse rate.

[Figure]

*Figure R2: Ratio between recovery time and plateau time for the full factorial parameter space. The numbers in the tiles show the values in %. The larger the ratio, the longer the ice sheet is in a partly recovered state and the shorter the temporary ice losses.*

Relatedly, Figure 2 shows that some of the parameter combinations do, apparently, have quite regular periods (especially in Figure 2b), while others do not (such as the higher lapse rates in Figure 2a). A short presentation of the values of the periods (and which are significant) should be done. The significant period values (kyr) could even simply be written inside the cyan blocks of Figure 4.

Thank you very much for the suggestion, we have adjusted the figure accordingly. Concerning the significance: we only identify oscillating states with a minimal amplitude of 0.5m as oscillation dynamics. We have clarified this in the text. In addition we tried using different weights for the average volume depicted in Fig. R1. Shifting it 25% closer to the minimal or maximal volume changed the computed oscillation times by less than 2.5% (less than 1% in most cases), and therefore the uncertainty is less than the variation in period between two oscillations periods.

As alluded to in my summary, I suggest replacing "oscillation" throughout the manuscript with a similar word that does not imply regularity, such as "fluctuation" or even "variation". This is because the sequence of states does not always have a regular repeat interval.

While we understand that the original term suggests a more regular oscillation than we observe, the term "fluctuation" suggests a much more random behavior than we observe. Although the term "variation" might offer some middle ground, we will stick to the original notation, as also suggested by the editor.

In addition the trajectories projected to the bedrock altitude vs. accumulation/ablation plane of the high-dimensional phase space, as shown in Fig. 6 (now Fig. 7 of the revised manuscript), form closed loops in phase space and therefore indicate that regular oscillations appear.

The first six lines of the Discussion restate the results, as do lines 11-17 on this page. These are redundant to the rest of the manuscript and should be removed. The last three lines of the first paragraph describe one possible extended importance of this study, which is not actually studied or discussed, and therefore would be more appropriate in the Conclusion or Introduction.

Thank you for the suggestions, we adapted the manuscript accordingly.

Finally, I would suggest a different name than "recovery" for the state where the ice sheet reaches a new equilibrium size significantly smaller than its start. "Recovery" implies, to me, that the ice sheet returns to its initial state. More precise names could be "re-equilibration" or "new steady state".

Many thanks for bringing our attention to the fact that "recovery" might imply that the Greenland Ice Sheet returns to its initial steady state. We chose this term because it seemed to suggest that the ice volume partly grows back after an initial ice loss. As both suggested terms do not transport this notion, which seems important in the context of the paper, we would rather stick with the term "incomplete recovery". This should not imply that the full initial ice volume is recovered, but it still tells the story of the regrowth.

**Technical corrections:**

P1 L5 - "Greenland could become essentially ice-free on the long-term" - I suggest stating the rough number of years found for this, instead of the vague "long-term".

Done

P1 L13 - "oscillation periods of tens to hundreds of thousand of years" - similarly, I suggest stating the rough number of years here. This is because your minimum period (80 kyr) is not that well described by "tens of thousand of years", so it is unintentionally misleading.

Done

P4 L4 - add Laurentide Ice Sheet, which is what Bassis et al. (2017) studied.

Done

P5 L8 - Please include a brief explanation, and/or citation, for why the enhancement factors (1 and 1.5) are different depending on which stress balance is used across the domain.

We included the reference to Ma et al. (2010), as they suggest using different enhancement factors in different flow regimes to reflect the anisotropy of the ice. However, the ratio between the enhancement factors used in our simulations is much less than the suggested ratio of 5-10, it is rather a result of an optimization process, similarly to Aschwanden et al. (2016), which we now also cite in this context, and very much depends on the model resolution.

P5 Sect 2.1 - The level of description of the ice sheet model (2.1.1) is much more general than the earth deformation model (2.1.2). The classic bending-beam PDE (Eq.1) is included with all parameters described and values supplied, for instance, but the sliding law and till stress model used in PISM are only described in words, with no parameter values given. These should be enhanced to match the level of 2.1.2.

Thank you very much for the suggestion. We have improved on the level of detail for the ice sheet modeling part.

P8 L12 - Missing reference (?).

Fixed it

P11 L3 - specify meters global sea level rise; write $1 \times 10^{19}$ instead of 1e+19

Done

P11 L21 - typo "2astern"

Fixed it

P12 L4 - I have never seen a zero-indexed "o/i/ii/iii" list before. I suggest standardizing to "i/ii/iii/iv".

Done

Table 1 - Specific values used for $\Delta T$ are listed, which is helpful. Values for $\Gamma$ and $\nu$, instead of their ranges, should be listed similarly.

Done

Figure 2 - Title of panel a is missing the "times" sign. X axis labels in kyr would make it more legible.

Done

Figure 5 - I suggest you outline or stipple the boxes that you classify as oscillating. As it is, the figure relies on the reader to interpret on their own which boxes show "significant difference" in color.

Thank you for the suggestion, we have now highlighted the region where oscillation dynamics take place.

Figure 6 - What is the mantle viscosity & climate change forcing ($\Delta T$) used here? It looks like it might be the same runs shown on Figure 2a, but that is only my guess.

Done

**Anonymous Referee #2**

Referee comment on "Dynamic regimes of the Greenland Ice Sheet emerging from interacting melt-elevation and glacial isostatic adjustment feedbacks" by Maria Zeitz et al., Earth Syst. Dynam. Discuss., https://doi.org/10.5194/esd-2021-100-RC2, 2022

**General comments**

The paper documents intriguing dynamic behaviour of the Greenland ice sheet resulting from the interplay between the melt-elevation feedback and the GIA feedback. The material is generally well presented and easy to follow. By itself the results are very interesting and potentially provide a very novel insight into the longer-term internal dynamics of the coupled climate-ice sheet-bedrock system. At the same time, I am also very puzzled by the results, in particular by the self-sustained quasi-periodic oscillations the authors find for (a rather narrow range) of parameter combinations. Many Greenland ice sheet modelers have performed similar experiments already since the early 1990s by imposing a stepwise warming in very similar model setups involving quasi the same degree-day type of climate forcing and taking into account isostasy with state-of-the-art models, but none of these studies have ever found even a trace of the kind of oscillations described in the paper. This makes me conclude that indeed their oscillatory behaviour may well be an 'artifact' (to cite their own words) of their particular experimental design and parameter choice. In other words, their model behaviour is probably not a very robust type of behaviour, to say the least, and might be

very difficult to replicate in other models. My suspicion is that their model behaviour is a result from the particular choice of the Lingle-Clarke isostatic model and will not show up for any other isostatic model, be it of the 'ELRA' type, or of the more sophisticated 'self-gravitating visco-elastic earth model' type.

I think the paper would be a very valuable addition to the literature of Greenland ice sheet dynamics, but first I would like to find out more on the robustness of the results and the specific role played by isostasy. A particular feature of the Lingle-Clarke model, and its implementation by Bueler et al. (2007) is that the relaxation time increases for wavelengths up to a few thousand km (a wavelength corresponding to the Greenland situation), which I believe is unrealistic. Full visco-elastic models show the contrary, the relaxation time decreases for a larger load. My guess is that it is exactly this specific behaviour of the LC model that is causing the oscillations. I suggest the authors make an effort to respond to this criticism by including material (figures and/or discussion) to prove or disprove this point.

Thank you very much for this comment. As the oscillation dynamics can be interpreted as a result of two competing feedbacks, as described in Fig. 1 and Fig. 4 of the manuscript, which manifests over a wide range of parameters and modeling choices (see discussion below), we do not think that it is an 'artifact' of this particular modeling setup alone. Talking to other ice-sheet modelers revealed that experiments with a constant climate forcing on time scales of many hundreds of millenia are performed less often compared to e.g. paleo climate forcing or sea-level rise experiments on shorter time scales. Therefore it may be less surprising that this dynamic regime has not been reported in the literature yet.

In order to test if the oscillatory behavior is an artifact of the bedrock model alone, we tested some experiments with the already implemented point-wise isostasy model, an instantaneous pointwise adaption of the bedrock to changes in load

$$b(t,x,y) = b(0,x,y) - \frac{\rho_i}{\rho_m}[H(t,x,y) - H(0,x,y)].$$

Here $b(t,x,y)$ is the elevation of the bedrock at a given time and location, $H(t,x,y)$ is the ice thickness at a given time and location and $\rho_i$ and $\rho_m$ are the density of the ice and the mantle, respectively. Simulations with this most simple bedrock model do show very similar oscillations, even though the amplitude and the oscillation time differ from the oscillation which appear with the Lingle-Clark model, see Fig. R3 a. We show and discuss these results in a new subsection dealing with robustness towards several modeling choices.

[Figure]

*Figure R3: Robustness analysis for the simulation run with parameters $\Delta T = 2K$, $\Gamma = 6K/km$, $\eta = 1 \times 10^{21}$Pa s. The gray curve shown in each panel corresponds to the reference run as described in the paper. Robustness tests are shown in color with faint, thin lines. One curve is highlighted in each panel to increase readability. (a) comparison to the pointwise isostacy model, described in the equation above. (b) comparison to a run, which includes a precipitation increase of 7.3% for every one degree Celsius of global mean temperature increase. (c) comparison of different spin-ups. The green curve has been spun up with two glacial cycles, while the gray curve was spun with a constant climate. (d) comparison to a run without flux correction. (e) comparison with different lithosphere thicknesses. The lithosphere flexural rigidity was varied corresponding to two different effective lithosphere thicknesses of 50km and 100km. The reference run assumes a lithosphere thickness of 88km. (f) comparison with a different interpolation for the transition between rain and snow.*

Moreover, a discussion with Michele Petrini revealed that he indeed observes long-term oscillations with a constant climate forcing using the ELRA bedrock model and a lapse rate of 6K/km. The peer-reviewed publication showing these results is forthcoming; a first glimpse can be found in the display materials of the 2021 EGU contribution.

In addition to the changes in amplitude and oscillation time, which can be seen above, a shift in the dynamic landscape (Fig. 4) might be a consequence of alternative modeling choices. For example when combining the glacial spin-up with the no flux correction (see panels c and d), the response of the Greenland Ice Sheet changes from "oscillation" to "partial recovery". Performing new simulations for the full factorial parameter space for each possible combination of modeling choices is sadly beyond the scope of this paper.

Having performed a suite of robustness experiments with different modeling choices, we still find the qualitatively similar oscillating behavior. We therefore conclude that the oscillating regime is fairly robust and not an artifact driven by the choice of the solid Earth model alone.

**Specific comments**

page 2, line 5: a reference is needed to substantiate the 65/35% attribution of current ice losses of the Greenland ice sheet. As far as I am aware from comprehensive studies, the ratio is more like 50/50 for both SMB changes and ice calving changes (e.g . IMBIE team, 2020)

Thank you for the comment. We have added the IMBIE reference and adjusted the numbers. The original numbers were taken from Mouginot et al. (2019) and it is an average value over the period from 1972 - 2018. However, with increasing warming the changes in climatic mass balance have become more important (Mouginot et al. (2019) find 55% for the period 2000-2018).

page 2, line 22: here, and elsewhere (page 6, line 5) it is stated that 'to our knowledge' their have been no previous studies coupling Greenland ice-sheet dynamics to bedrock dynamics. That is not entirely true. Le Meur and Huybrechts (1998, also in GJI in 2001) have done this for the glacial cycles, also in Zweck and Huybrechts (2005) Greenland ice sheet dynamics was included and was part of the sensitivity study.

Thank you very much for bringing our attention to this. We have included these references to the manuscript and have clarified the sentence to "However, the interaction of the negative bedrock uplift feedback and the melt-elevation feedback, has, to our knowledge, not yet been systematically studied in the context of the Greenland Ice Sheet".

page 4, line 11: explain what the 'small ice cap instability' is.

Done

page 4, line 12: To what does 'This' refer?

We have clarified this sentence.

page 4, line 17: explain why the factor 1/3 is expected.

The ⅓ stems from the difference in densities. As the asthenosphere is approximately three times more dense than the ice, Archimedes' principle allows us to estimate the amount of uplift after a change in ice load. We have included this in the manuscript.

page 6, section 2.1.2: a critical appraisal of the specific features of the LC model is in order here. A more thorough discussion of the dependence of the relaxation time on the wavelength of the load change and how this compares to other models is required here, as this may well be a crucial issue in this paper.

The widely used ELRA model assumes one single relaxation time for the solid Earth response, independent of the wavelength of the load change, the Lingle Clark model in contrast includes the viscosity of the mantle explicitly, and therefore the relaxation time depends on the wavelength of the load change. Some discussion is already performed in Bueler et al. (2007), but we now show how the relaxation time vs. wavelength of the load change depends on mantle viscosity in the appendix of the revised manuscript.

[Figure]

*Figure R4: Relaxation times of the Lingle-Clark model vs. wavelength of the load change. Relaxation times are computed as in Bueler et al. (2007), Eq. (14).*

In contrast to more complex solid Earth models, the Lingle-Clark model exhibits only one single mode of the spectrum, the mantle mode M0. For a two layer model with a compressible elastic lithosphere over an viscous half-space with $\eta = 1 \times 10^{21}$ Pa s, the M0 mode has a maximal relaxation time of 10,000 years for a wavelength of approx. 300 km (see Klemann, 2003), the LC model represents this behavior well. Solid Earth models which include more layers exhibit also more modes of the spectrum. The additional modes in e.g. a four-layer model show a monotonous strong increase of relaxation time with increasing wavelength and decreasing wavenumber (Klemann 2003).

We now include a more detailed discussion of these features in the revised manuscript. However, we do have reason to believe, that the oscillations are not a feature of the LC model in particular, as they also arise when using the pointwise isostatic model (as shown in the robustness analysis), which has an instantaneous response time independent of the wavelength, as well as for the ELRA model as studied bei Petrini et al..

page 6, section 2.1.3: apparently the precipitation pattern from RACMO does not interact with climate change or ice sheet geometry as it seems to be fixed. Mention this explicitly and mention the shortcomings of such an approach.

We now mention the fact that the precipitation does not scale with temperature more explicitly and we briefly discuss that this makes the experiments less realistic. However, this idealized approach brings us a bit closer to the as well idealized feedback loop in Figure 1. We now also show in the robustness analysis that the simulations, which increase the precipitation by 7% per degree of warming show the same qualitative behavior in the oscillating regime, however, the amplitude of the oscillation is dampened by the increase in accumulation.

page 7, section 2.1.3: is the rain fraction a function of the monthly mean temperature? If so, the transition temperature range between 0 and 2°C seems much too small. One would still expect rainfall during a month with a mean temperature below 0°C and snowfall for a mean temperature above 2°C. Please discuss the limitations of this approach.

PISM is a stand-alone ice sheet model, which relies on input for atmospheric variables like precipitation or near surface air temperature. The interpolation described here is the standard for how PISM treats precipitation and it allows change between snow and rain depending on the temperature. A linear transition between rain and snowfall is a very simple approximation of more complex processes which would of course be best described with a fully coupled atmospheric model. We now include an additional run, which has a broader transition range, from -1°C to +3°C, which also shows oscillating behavior, but with a higher amplitude (see Fig. R3 f).

page 7, line 16: it is mentioned that ice-ocean interaction is included via PICO. More information is needed here. Where is the ocean forcing coming from? At what resolution? What about water circulation in the fjords? How is oceanic forcing transferred to calving fronts? Does the model have a grounding line and attached ice shelves, and how are they treated? Does it matter to include ocean forcing for the type of experiments described here at all?

We generate the ocean forcing ourselves by using a scalar anomaly on the World Ocean Atlas data. The ocean warming corresponds to 70% of the global mean temperature anomaly. The WOA data is remapped onto the PISM simulation grid. The data is averaged over the extended drainage basins of the GrIS. Here the average value is applied for each extended drainage basin, even if the ice sheet retracts. PICO calculates the sub-shelf melt rate; it is not suited to compute the plume-driven frontal melt of a tidewater glacier without a floating tongue. The calving process is computed through von Mises calving and a minimal floating thickness of 50m.

It is true that the ocean forcing might not be necessary for this kind of experiment, as the floating ice tongues make up less than 0.2% of the ice sheet area.

page 7, section 2.2.1, and associated figures in the supplement: it is puzzling to me that while the climatic mass balance from the model differs substantially from RACMO (Fig. S2), the simulated ice sheet domain almost exactly matches the observations (Figs. S1 and S3). Almost on view it can be seen that the ice-sheet wide average surface mass balance must be positive over the domains shown, yet there is hardly any advance of the margin for the initial state. How was the initial ice sheet constrained? What is the meaning of the row of black points (low or zero velocity) at the margin in Fig. S3? To me it is hard to believe that the initial state corresponds to a self-sustained steady-state ice sheet with a freely evolving margin, the latter of which is crucial in the experiments.

Yes, it is correct that we used a flux correction for the zones, which are ice-free under the present day climate. Ice sheet models forced with the precipitation fields from RCMs often overestimate the accumulation in the South East and therefore an ice sheet in present-day climate would grow in volume. As we perform warming experiments, where the ice sheet experiences retreat, we do not think that including such a flux correction should affect the results on long time scales. We now provide a simulation run, which does not use the flux correction on the ice-free margin and we still observe oscillations, which is found in the robustness analysis section.

A consequence of removing the mask is that the control run stabilizes at slightly higher ice volumes (from 7.06m to 7.62m).

We have included a better discussion of this in the paper.

page 12 and further, section 3.2: A crucial issue is how realistic the bedrock model is. In the model only viscosities are changed to control the relaxation time scale. What about the effect of variations in flexural rigidity of the lithosphere?

We have performed runs with two additional elastic lithosphere thicknesses, which all show qualitatively similar behavior. It is noticeable though, that thicker lithospheres show a stronger initial ice loss than thinner lithospheres. Both additional simulations are similar in amplitude and oscillation time. Note that the LC model does not include lithosphere thickness directly but through the flexural rigidity.

page 14, figure 6: The figure is very difficult to read and understand, and should be improved. The colour saturation seems to represent time (but the caption does not say), however the pale parts of the lines are difficult to see. What is the meaning of both crosses? Lower axis: accumumlation-> accumulation. Left axis: are you sure the average level of topography has negative values? Please adapt the figure and the caption to increase readability.

Thank you very much for the comment. We noticed an error in the python script used for the analysis and now fixed the mistake which led to negative average bedrock topography values. In addition we have improved the visual readability of the figure, by adjusting the color scale and adding the initial states (which is the meaning of the crosses) to the legend. The caption is now also improved.

In our opinion the figure, even if a bit unusual in the ice-sheet modeling context, provides a nice visual representation of the oscillatory behavior. The trajectory of the oscillation dynamics forms closed loops in phase space (or rather when projected to this plane of the high-dimensional phase space), similar to a non-linear oscillator or a limit-cycle.

We have shortened the discussion of the figure and removed the last sentence of the paragraph, as the correct trajectories now intersect.

[Figure]

*Figure R4: Corrected trajectories projected onto one plane of the phase space. Compare with Fig. 6 in manuscript.*

Page 15, lines 18-20: it is impossible to discern on Figure 6 the clockwise or counterclockwise sense of the trajectories. Perhaps an arrow would help.

Thank you for the suggestion, we have included arrows in the revised manuscript.

Page 16, line 31: Petrini et al. (2021) is a crucial reference to prove that the results are not an artifact of the specific experimental design. However, that is an EGU abstract, and cannot be checked. Remove the reference to Petrini et al. (2021).

The robustness tests we have included now should provide sufficient evidence that the observed behavior is most likely not an artifact, and almost certainly not an artifact of the LC model.

Petrini et al. observed oscillations in long-term warming runs with the CISM ice sheet model coupled to an ELRA model, when forced with a constant climate from CESM SMB. Those oscillations are regular with an approximate period of 30-40kyears. While the abstract alone is not a good enough reference, the interested reader will find the "display materials" showing the time series of the above-mentioned runs, linked to the abstract (https://presentations.copernicus.org/EGU21/EGU21-12958_presentation.pdf). Instead of citing the EGU contribution we could cite the display materials directly as a web page, if this is more suitable. This should provide a first impression to the reader, while the peer reviewed scientific publication is being prepared by Petrini and his co-authors.

Technical corrections

Page 3, line 14: solte -> solve
Page 3, line 32: sophisticates -> sophisticated
Page 3, line 33: year of Fettweis et al. publication missing
Page 4, line16: add 'itself' between 'manifest' and 'on'
Page 8, table 1: the mantle viscosity value of 1x1**-19 cannot be right.
Page 8, line 12: there is a '?' in the reference list
Page 8, line 22: remove the comma between 'both' and 'the'.
Page 14: line 11: do not start a sentence with a capital after a semi-colon
Page 16, line 31: Hoever -> However

We have corrected the manuscript accordingly.

**Comment on esd-2021-100**

**Michel Crucifix (Editor)**

Dear authors,

After a (somewhat) long wait, the two reviews are in. What you show in your paper can, in my view, be called 'oscilations' (even if not perfectly periodic), and oscillations in glacial/isostatic systems are rare, though not quite unprecedented (Oerlemans, J. Glacial cycles and ice-sheet modelling. Climate Change, **4,** 353–374 (1982). https://doi.org/10.1007/BF02423468. The context was quite different, as well at the overall setup, but this old example suggests, as pointed out by reviewer 2, that assumptions invoved in the lithosphere/asthenosphere model are crucial. I would therefore invite you to consider the possibility of sensitivity experiments that would adress the question,  though I would not be willing to substantially delay the publication of your study.

Many thanks for bringing our attention to the publication by Oerlemans, discussing free oscillations in a conceptual model including an ice sheet, a simplified melt-elevation feedback and a simple bedrock uplift model with one single relaxation time. In fact we believe that this paper strengthens our point, as it shows that unforced oscillations can emerge from the feedbacks involved in a variety of modeling assumptions, given that the system is "nonlinear enough" and includes thermodynamics. There the occurrence of oscillations depends on snow-line slope, the maximal accumulation rate and the representation of thermodynamics. The bedrock parameters were not part of the study. We now include a reference to this publication in the discussion.

[revised manuscript text omitted]

---

## Author Response (AR2)

**Response to Reviewer Comments**

Journal: Earth System Dynamics Title: **Dynamic regimes of the Greenland Ice Sheet emerging from interacting melt-elevation and glacial isostatic adjustment feedbacks** Authors: Maria Zeitz, Jan Haacker, Jonathan Donges, Torsten Albrecht and Ricarda Winkelmann MS No: esd-2021-100 MS Type: Research article

Once again, we would like to thank the editor Michel Crucifix and the reviewer, Kristin Poinar for her helpful comments! In our second minor revision of the manuscript we have addressed the suggestions made by the reviewer (point by point answers below). In addition we have reworked the color schemes of the paper to fit the Copernicus standard for accessibility concerning color blindness and have added the revised version of Figure 6, which should have included a visual help to highlight the oscillation region in parameter space already in the last revision (which does not change the data of the figure).

We provide detailed answers to all comments below. The reviewers' comments are given in black and the authors' in blue. The changes made to the manuscript can be found in the track-changes file (created with latexdiff).

**Suggestions for revision or reasons for rejection (will be published if the paper is accepted for final publication)**

**Kristin Poinar (Referee)**

Review of revised manuscript "Dynamic regimes of the Greenland Ice Sheet emerging from interacting melt-elevation and glacial isostatic adjustment feedbacks" by Maria Zeitz et al.

June 7, 2022

The manuscript has been improved and enhanced from the first submission with the addition of a robustness analysis (section 4.3) on the oscillations across six different modeling choices (e.g., bedrock uplift model, spin up state, lithosphere thickness). An appendix is also added that illustrates the dependence of the relaxation time of the Lingle-Clarke bedrock uplift model on the ice sheet length scale. Thirdly, the method for measuring the oscillation timescale is now explained (I believe it is simpler than the method used in the first submission), and the uncertainties on these timescales quantified at 1 - 2.5%.

The new figure that illustrates the ratio of the recovery time to the plateau time (new Figure 5) is a good addition. This shows the result that, for most parameter combinations, an oscillating ice sheet spends more time near its "big" state than around its "collapsed" state (these are termed the "plateau" and "recovery" states, respectively, per page 15). It took me a little time to find my bearings in Figure 5; I think this was due to the use of percentages in the text (e.g. "10%" on page 15) versus whole numbers without a % sign on the figure

(e.g. "14" for the corresponding 10% point on Figure 5, with the "%" appearing in the colorbar), and the use of ratios in the caption (i.e. "1" instead of "100%"). These should be made consistent in some way. (To be clear, I don't have any problem with calling 10% ~ 14%.)

Many thanks for this comment. We have harmonized the notation in the Figure, the caption and the corresponding paragraph. In addition we have decided to be more precise in the text as well and have changed from 10% to 14%.

The new Figure 5 and the new section 3.2.2 go toward addressing my question about why there is no clear pattern between parameter combinations and oscillation timescales. With their new definition of oscillation timescale, the authors have now found that an ice sheet spends more time in the collapsed / recovery state in model runs with higher temperatures, higher mantle viscosity, or higher lapse rate (page 15). If I think through, myself, about which direction I would expect these parameters to work in and why, I think I can agree with the results -- but more ideally, the authors would explain these findings themselves in the discussion. I did not find this in the manuscript. A short additional subsection in the discussion would help readers make more sense of this central finding.

We have added a subsection in the discussion, 4.2.1., explicitly addressing the interpretation of the findings following from Figure 5. In particular we explicitly state that a high fraction of "recovery time" / "plateau time" seems to indicate a loss of stability of the Greenland Ice Sheet in these simulations.

I also like the change to Figure 1. It now more clearly shows the signs of both feedbacks by highlighting the signs of each process, showing twin arrows along the common pathway, and the processes are now named more clearly.

**Thank you.**

I was not able to assess the other reviewer's comment, and the authors' response, regarding whether the wavelength-dependence of relaxation time in the Lingle-Clarke model was realistic. I do agree with the reviewer that if this is an artifact of Lingle-Clarke, this paper would be better framed as "Watch out for this behavior in your ISM if you're using Lingle-Clark" rather than "This is a true feature of the GrIS on long timecales". The authors responded to this query by testing other bedrock models, which produced similar results as Lingle-Clark, but I lack the expertise to assess whether those models have similar weaknesses to Lingle-Clarke, versus the "full visco-elastic" models requested by the reviewer.

Thank you for the effort to assess the reviewer's concerns.

Indeed, the interactive coupling of the Parallel Ice Sheet Model (PISM) to full visco-elastic GIA models is partly ongoing work. Torsten Albrecht, one of the authors of this paper, is working on interactively coupling PISM to the VIscoelastic Lithosphere and MAntle model (VILMA) (Klemann et al., 2008; Martinec et al., 2018) and focusing particularly on the Antarctic Ice Sheet. First results for the Greenland Ice Sheet show that indeed, the self-sustained oscillations are indeed reproducible in this setup, and we hint on that in the "robustness" section. However, a proper analysis of the dynamic regimes in the interactively coupled setup would go beyond the scope of the present manuscript: a proper introduction and validation of the coupling scheme are still missing and part of forthcoming work. Regardless of the outcome of the bedrock uplift model exercises, I find that the authors have improved the manuscript from its original version to make a stronger paper overall. With small changes to the Figure 5 description and a thorough discussion of what the Figure 5 findings mean, the concerns I had will all be addressed.

Cheers, Kristin Poinar